# Thermogenic hydrocarbon biodegradation by diverse depth-stratified microbial populations at a Scotian Basin cold seep

Xiyang Dong [1,2✉], Jayne E. Rattray [2], D. Calvin Campbell [3], Jamie Webb[4], Anirban Chakraborty [2], Oyeboade Adebayo [2], Stuart Matthews[2], Carmen Li[2], Martin Fowler[4], Natasha M. Morrison[5], Adam MacDonald[5], Ryan A. Groves[2], Ian A. Lewis[2], Scott H. Wang[2], Daisuke Mayumi[6], Chris Greening [7,8] & Casey R. J. Hubert [2✉]

At marine cold seeps, gaseous and liquid hydrocarbons migrate from deep subsurface origins to the sediment-water interface. Cold seep sediments are known to host taxonomically diverse microorganisms, but little is known about their metabolic potential and depth distribution in relation to hydrocarbon and electron acceptor availability. Here we combined geophysical, geochemical, metagenomic and metabolomic measurements to profile microbial activities at a newly discovered cold seep in the deep sea. Metagenomic profiling revealed compositional and functional differentiation between near-surface sediments and deeper subsurface layers. In both sulfate-rich and sulfate-depleted depths, various archaeal and bacterial community members are actively oxidizing thermogenic hydrocarbons anaerobically. Depth distributions of hydrocarbon-oxidizing archaea revealed that they are not necessarily associated with sulfate reduction, which is especially surprising for anaerobic ethane and butane oxidizers. Overall, these findings link subseafloor microbiomes to various biochemical mechanisms for the anaerobic degradation of deeply-sourced thermogenic hydrocarbons.

[1] School of Marine Sciences, Sun Yat-Sen University, Zhuhai, 519082, China. [2] Department of Biological Sciences, University of Calgary, Calgary, AB T2N 1N4, Canada. [3] Geological Survey of Canada-Atlantic, Dartmouth, NS B3B 1A6, Canada. [4] Applied Petroleum Technology (Canada), Calgary, AB T2N 1Z6, Canada. [5] Nova Scotia Department of Energy and Mines, Halifax, NS B2Y 4A2, Canada. [6] Institute for Geo-Resources and Environment, Geological Survey of Japan, National Institute of Advanced Industrial Science and Technology (AIST), 1-1-1 Higashi, Tsukuba, 305-8567, Japan. [7] School of Biological Sciences, Monash University, Clayton, VIC 3800, Australia. [8] Department of Microbiology, Biomedicine Discovery Institute, Monash University, Clayton, VIC 3800, Australia. ✉email: dongxy23@mail.sysu.edu.cn; chubert@ucalgary.ca

Marine cold seeps are characterized by the migration of gas and oil from deep subsurface sources to the sediment-water interface[1,2]. This seepage often contains gaseous short-chain alkanes, as well as heavier liquid alkanes and aromatic compounds[3], which originate from deep thermogenic petroleum deposits. Migrated hydrocarbons can serve as an abundant source of carbon and energy for microorganisms in these ecosystems, either via their direct utilization or indirectly through metabolizing by-products of hydrocarbon biodegradation[4]. Multiple 16S rRNA gene surveys have revealed that cold seep sediments at or near the sediment-water interface host an extensive diversity of archaeal and bacterial lineages[5–9]. However, much less is known about metabolic versatility of this diverse microbiome, and how surface and subsurface populations are distributed in different redox zones within the sediment column[10,11]. Most seep-associated microorganisms lack sequenced genomes, precluding meaningful predictions of relationships between microbial lineages and their biogeochemical functions[4,8,10–13].

Geochemical studies have provided evidence that microorganisms in deep seafloor sediments, including cold seeps, mediate anaerobic hydrocarbon oxidation[2,3]. A range of efforts have been undertaken to enrich and isolate anaerobic hydrocarbon-oxidizing microorganisms from cold seep sediments and other ecosystems rich in hydrocarbons (e.g., marine hydrothermal vents)[5–9,14]. Numerous studies have focused on the anaerobic oxidation of methane, as methane generally is the dominant hydrocarbon in cold seep fluids. This process is mediated by anaerobic methanotrophic (ANME) archaea through the reverse methanogenesis pathway, typically in syntrophy with bacteria that can reduce electron acceptors such as sulfate, nitrate, and metal oxides[8,15,16]. Investigations of enrichment cultures have also revealed anaerobic bacterial or archaeal oxidation of non-methane alkanes and aromatic hydrocarbons, including ethane (e.g., *Ca.* Argoarchaeum and *Ca.* Ethanoperedens)[17,18], *n*-butane and propane (e.g., *Ca.* Syntrophoarchaeum and *Desulfobacteraceae* BuS5)[9,19], dodecane (e.g., *Desulfosarcina/Desulfococcus* clade)[6], and naphthalene (e.g., deltaproteobacterial strain NaphS2)[20,21]. Alkane-oxidizing archaea normally do so in consortia with sulfate-reducing bacteria whereas bacteria known to degrade hydrocarbons usually couple this to sulfate reduction in a single-cell process[8,22]. Long-chain alkanes can also be metabolized in syntrophic partnerships, e.g., by bacteria in the genera *Smithella* and *Syntrophus*, together with methanogenic archaea[23,24].

Despite this progress, it remains uncertain whether anaerobic hydrocarbon-degrading isolates or consortia studied in enrichment cultures play these roles in situ in deep sea sediments. Thus, metabolic functions of microbial communities have also been assessed using cultivation-independent approaches including sequencing of phylogenetic and/or functional marker genes and environmental metagenomics. Single-gene surveys, for example, investigating the diversity of genes encoding enzymes for alkane or aromatic compound activation via addition to fumarate[3,11,25], have increased our knowledge about the phylogenetic diversity of hydrocarbon degraders. Most genome-resolved metagenomic studies have focused on hydrothermally influenced sediments that are rich in hydrocarbons, e.g., Guaymas Basin in the Gulf of California[26–28]. These studies have provided insights into the phylogenetic diversity and functional capabilities of potential hydrocarbon-degrading microorganisms, including the discovery of *Ca.* Helarchaeota from the Asgard superphylum with the potential for hydrocarbon oxidation using methyl-CoM reductase-like enzymes[28]. However, studies integrating geochemical processes and microbial metabolism in redox-stratified deep sea sediments are lacking.

In contrast to hydrothermal sediments, there have been fewer reports on the metabolism of hydrocarbons and other compounds in cold seep sediments, especially in the deep sea[4]. One of the best-studied cold seep areas is the Gulf of Mexico where deeply sourced hydrocarbons rise through continental slope sediments fractured by salt tectonics[8,13,29]. Compared with hydrocarbon seep ecosystems along active margins and in petroleum rich basins like the Gulf of Mexico, much less is known about those along passive margins[30]. The Scotian Basin is at the volcanic and non-volcanic transition continental margin, extending over an area of ~260,000 km² in the northwest Atlantic Ocean, offshore Nova Scotia in eastern Canada (Fig. 1). Based on satellite and seismic reflection data, this area shows strong evidence for seepage of thermogenic hydrocarbons with occurrences of high-pressure diapirs, polygonal faults, pockmarks, and gas chimneys[31].

In this study, we combine geophysical, geochemical, and metabolomic analyses with gene- and genome-centric metagenomics to understand the communities and processes responsible for anaerobic oxidation of different hydrocarbons, as well as their depth distributions, at a newly discovered deep sea cold seep caused by salt tectonics in this area. Through this work, we provide strong evidence supporting that (i) hydrocarbons are thermogenic and experience biodegradation upon migration up into surface sediment layers; (ii) these processes are actively performed in the cold deep sea by bacteria and archaea through diverse biochemical mechanisms; and (iii) the microbiome catalysing anaerobic hydrocarbon degradation at different depths is dependent on metabolic adaptations for different redox regimes.

## Results

**Migrated thermogenic hydrocarbons are subject to biodegradation.** At a water depth of 2306 m, a 3.44-meter-long piston core was retrieved from the Scotian Slope, off the coast of Eastern Canada (Fig. 1a). The 3D seismic survey indicated that this site is located above a buried salt diapir (Fig. 1b). An overlying seismic amplitude anomaly was interpreted to be a direct hydrocarbon indicator with salt diapir-associated crestal faults suggestive of a potential conduit for fluid migration to the seafloor. In the bottom of the core, at 332–344 cm below the seafloor (cmbsf), gas hydrates were observed in frozen crystalized form and numerous gas bubbles escaped during retrieval. A strong sulfide odor was also detected during core retrieval and processing. Molecular and isotopic compositions of two headspace gas samples subsampled from the sediments adjacent to the gas hydrates (332–337 and 337–344 cmbsf) detected 7446 and 4029 ppm of total hydrocarbon gases (THG), respectively, primarily made up of methane (85% and 79%) with considerable proportions of $C_2$-$C_4$ gases (3.22% and 6.54%) (Table 1). In order to assess the origin of the hydrocarbon gases, their molecular and isotopic compositions were compared to recently revised genetic diagrams[32], focusing on $\delta^{13}C$-$C_1$, $C_1/(C_2 + C_3)$, $\delta^2H$-$C_1$, $\delta^{13}C$-$C_1$, and $\delta^{13}C$-$CO_2$. All measured geochemical parameters are within the range defined for gases of thermogenic origin indicating that they migrated upward from a mature petroleum source rock (Table 1). Ethane and propane were $^{13}C$-enriched compared with methane, likely reflecting the addition of some biogenic methane to the migrating thermogenic gas, as well as biodegradation of ethane and propane[32]. Ratios of *iso*-butane to *n*-butane were 1.6–1.8 and suggest preferential consumption of more labile *n*-alkanes[3].

Sediments from four additional depths were analyzed for extractable organic matter (EOM, i.e., $C_{12+}$ hydrocarbons), showing high yields (104–361 mg/kg rock) comprising saturated hydrocarbons (25–52%), aromatic hydrocarbons (10–14%), and

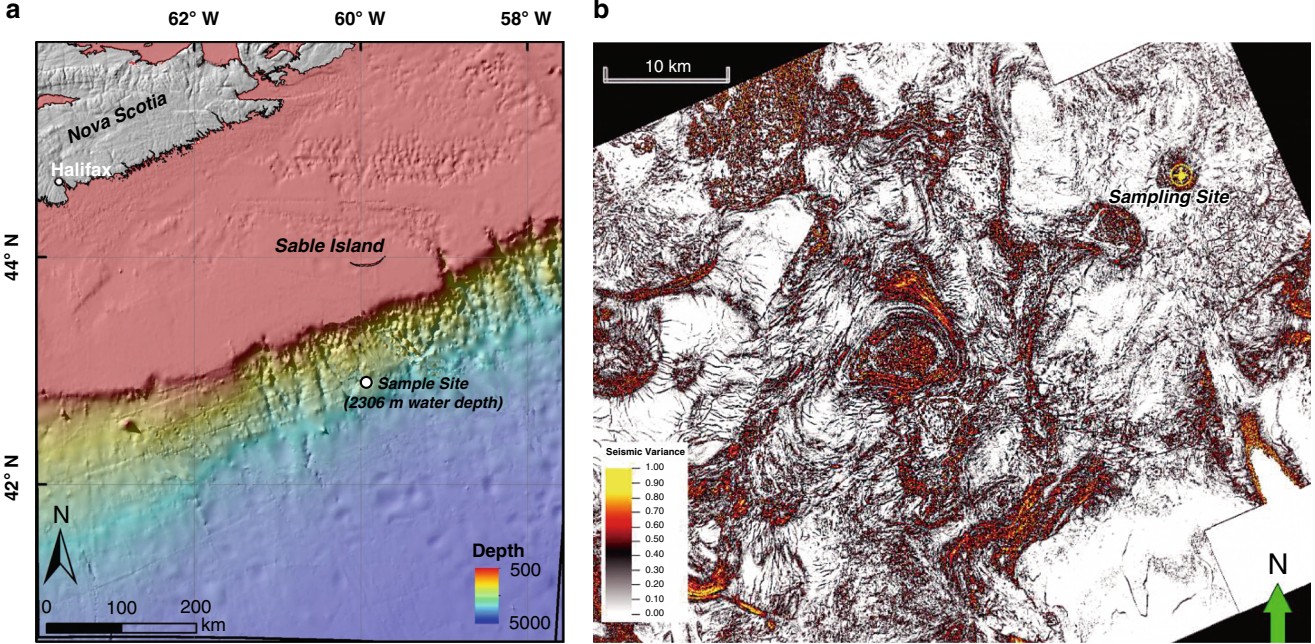

**Fig. 1 Bathymetry and 3D seismic characterization of the study area. a** Bathymetry of the Scotian Basin and the sediment sampling location. The Scotian Basin is in the northwest Atlantic Ocean, offshore Nova Scotia in eastern Canada. A 3.44-meter-long piston core was retrieved at a water depth of 2306 m. The map was generated using the ESRI ArcMap v10.6.1 software. The background bathymetry data were retrieved from General Bathymetric Chart of the Oceans (GEBCO) (https://www.gebco.net/data_and_products/gridded_bathymetry_data/documents/gebco_08.pdf). **b** Horizontal time slice through the Tangier 3D seismic survey at −3500 m depth (i.e., approx. 1200 mbsf) with the variance (coherence) attribute shows salt diapirism throughout the Scotian Basin, including at the site chosen for the sediment core. Drawn with the Schlumberger Petrel v2018.1 software.

**Table 1 Molecular and isotopic compositions of two gas samples sourced from sediments subsampled from the core regions nearest gas hydrates.**

| Depth (cmbsf) | | 332–337 | 337–344 |
|---|---|---|---|
| Features | | Bubbling gas | Gas hydrate |
| THG compositions (%) | Methane | 85.10 | 79.00 |
| | Ethane | 2.00 | 1.38 |
| | Propane | 3.10 | 1.25 |
| | *iso*-butane | 0.93 | 0.36 |
| | *n*-butane | 0.51 | 0.23 |
| | $\Sigma C_2$–$C_4$ | 6.54 | 3.22 |
| | THG (ppm) | 7446 | 4029 |
| | $C_1/(C_2+C_3)$ | 16.7 | 30.0 |
| Isotopic signatures (‰) | $C_1\ \delta^{13}C$ | −42.2 | −49.0 |
| | $C_1\ \delta^2H$ | −169 | −177 |
| | $C_2\ \delta^{13}C$ | −24 | −26 |
| | $C_3\ \delta^{13}C$ | −21.5 | −22.3 |
| | $i$-$C_4\ \delta^{13}C$ | −22.6 | ND |
| | $n$-$C_4\ \delta^{13}C$ | −21.6 | ND |
| | $CO_2\ \delta^{13}C$ | −6.3 | −7.7 |

$C_1$ methane, $C_2$ ethane, $C_3$ propane, $C_4$ butane. THG, total hydrocarbon gas. $C_1/(C_2+C_3)$, molecular ratios of methane to ethane and propane. $\delta^{13}C$ and $\delta^2H$, stable carbon and hydrogen isotope compositions. ND, not determined.

other components (Supplementary Table 1). Further gas chromatographic analysis of these oil-laden sediments revealed large unresolved complex mixture humps in the $C_{13}$–$C_{20}$ *n*-alkane elution range (Supplementary Fig. 1), indicative of hydrocarbon biodegradation[3]. Pristane and phytane, which are widely used internal conserved markers for oil biodegradation[33], were more abundant than $C_{17}$ and $C_{18}$ *n*-alkanes, suggesting preferential biodegradation of *n*-alkanes. Consistent with this, carbon dioxide was isotopically heavy in these sediments (Table 1) and clearly associated with secondary microbial degradation[32].

**Microbial communities are depth-stratified and metabolically diverse.** Deep shotgun metagenome sequencing was performed for sediments based on porewater sulfate concentrations and other geochemical data (Table 2 and Supplementary Fig. 1). Alpha diversity was calculated using single-copy marker genes from the metagenomic datasets[34], and cell densities were estimated via quantitative PCR of bacterial and archaeal 16S rRNA genes. Differences in diversity and cell density were related to the availability of migrated thermogenic hydrocarbons and sulfate (Table 1 and Supplementary Table 1). Overall, the surface

**Table 2 Key features of sediment samples used for microbiological analyses.**

| Samples | Depth (cmbsf) | Biogeochemical zone | Filtered paired reads | Shannon index* | Bacterial 16S rRNA (×10$^9$)** | Archaeal 16S rRNA (×10$^9$)** |
|---|---|---|---|---|---|---|
| S1 | 0 | Sulfate-rich | 61,241,391 | 6.70 ± 0.04 | 2.64 | 0.113 |
| S2 | 20 | Sulfate-rich | 59,469,560 | 5.58 ± 0.10 | 2.22 | 0.052 |
| S3 | 60 | Sulfate-rich | 57,920,960 | 3.60 ± 0.23 | 1.57 | 1.230 |
| S4 | 100 | Sulfate-depleted | 54,671,031 | 5.05 ± 0.13 | 0.13 | 0.010 |
| S5 | 150 | Sulfate-depleted | 53,856,875 | 4.97 ± 0.10 | 0.14 | 0.022 |
| S6 | 200 | Sulfate-depleted | 95,943,312 | 5.36 ± 0.08 | 0.06 | 0.004 |
| S7 | 250 | Sulfate-depleted | 70,775,877 | 4.99 ± 0.12 | 0.23 | 0.016 |

*Shannon index was calculated from metagenomes using 14 single-copy marker genes.
**Bacterial and archaeal numbers were estimated by qPCR, expressed as 16S rRNA genes per g sediment.

sediment and near-surface sediment (20 cmbsf) harbored the most diverse communities (Shannon index = 6.70 and 5.58) and highest bacterial cell density (2.64 and 2.22 × 10$^9$ 16S rRNA genes g$^{-1}$). In contrast, the sediment at 60 cmbsf harbored the most distinct communities (Fig. 2a), the lowest microbial diversity (3.60), and the highest archaeal cell density (1.23 × 10$^9$ 16S rRNA genes g$^{-1}$). Deeper sediments (100–250 cmbsf) were compositionally similar to each other and harbored moderately to highly diverse and abundant communities (Table 2 and Fig. 2a).

For taxonomic profiling, the phyloFlash pipeline[35] was applied to reconstruct 16S rRNA gene fragments from metagenome raw reads for each sediment depth. Dominant bacterial lineages in surface sediments were *Gammaproteobacteria* (21%), *Deltaproteobacteria* (14%), *Alphaproteobacteria* (11%), and Planctomycetes (14%), whereas Atribacteria (22–51%), Chloroflexi (5–32%), and *Deltaproteobacteria* (5–11%) were predominant at and below 20 cmbsf (Fig. 2a). Thaumarchaeota (mainly class *Nitrososphaeria*) were the most dominant archaea (92%) found in surface sediment, but were in low abundance in subsurface sediments (Fig. 2a). *Methanomicrobia* (Euryarchaeota) and Lokiarchaeota (Asgard group) were predominant at and below 20 cmbsf. ANME-1 (*Methanomicrobia*) comprised 97% of archaea at 60 cmbsf, consistent with the presence of its previously observed syntrophic partner SEEP-SRB1 bacteria[12], suggesting that this depth was part of the sulfate-methane transition zone (Fig. 2b, c). Taxonomic profiles produced by 16S rRNA gene amplicon sequencing were broadly similar to metagenomic profiling, but with differences in the relative abundances of specific groups (e.g., Chloroflexi and Heimdallarchaeota) (Fig. 2a and Supplementary Data 1). Together, both methods show that microbial communities throughout the sediment column are diverse and consist of mostly uncultured taxonomic groups.

Metagenomes were assembled for sediments from individual depths, and a co-assembly was performed by combining metagenomes from all depths[36] (Supplementary Data 2 and 3). Binning of derived assemblies was based on tetranucleotide frequencies and coverage profiles using several algorithms[37]. This analysis yielded 376 unique metagenome-assembled genomes (MAGs) with <99% average nucleotide identity from each other, having >50% completeness and <10% contamination based on CheckM analysis[38]. Recovered MAGs (293 bacterial and 83 archaeal) spanned 43 different phyla, most of which are poorly characterized without cultured representatives (Supplementary Fig. 2; Supplementary Data 2 and 4). Ten of these genomes could not be classified due to lack of reference genomes and, based on their phylogeny, might belong to five new candidate phyla (Supplementary Fig. 2). Several MAGs were affiliated with the Class *Methanomicrobia* ($n = 24$) within the phylum Euryarchaeota, including 12 MAGs belonging to ANME-1 and ANME-2 lineages. Bacterial MAGs were mostly represented by Chloroflexi ($n = 93$), Planctomycetes

($n = 32$), and *Deltaproteobacteria* ($n = 29$). Overall, the 376 MAGs captured the prevalent bacterial and archaeal lineages revealed by 16S rRNA gene analysis, which represented 63.3–90.6% of the genera present in metagenomes for the deeper 20–250 cmbsf (cf. only 12.2% for 0 cmbsf).

We further linked the structure of microbial communities to their metabolic capabilities in carbon acquisition and energy conservation strategies. Metabolic reconstructions of 376 MAGs revealed versatile catabolic capabilities for assimilating carbohydrates, peptides and short-chain lipids (Fig. 3a and Supplementary Data 5–8). Also widespread is the capacity to conserve energy and fix carbon using electrons derived from inorganic compounds such as sulfide, thiosulfate, ammonia, nitrite, carbon monoxide, and hydrogen (Supplementary Figs. 3–12). For electron acceptors, while microorganisms capable of respiring sulfate, oxygen, nitrate, and nitrite were detected, these capacities were not prevalent (Supplementary Data 5). Many genomes encoded putative reductive dehalogenase genes for organohalide respiration (Supplementary Fig. 13 and Supplementary Data 9). In line with other recent observations[4], fermentation appears to be a universal strategy in these sediments, with production of hydrogen, formate, and acetate as major end products (Supplementary Data 5); accordingly, the majority of the recovered MAGs encoded bidirectional [NiFe]-hydrogenases (Fig. 3a). Supplementary Note 1 further elaborates upon the broader metabolic capabilities of the community. The sections below focused on using gene- and genome-centric approaches to identify novel microorganisms and specific catabolic pathways involved in anaerobic degradation of deeply sourced thermogenic hydrocarbons. Metabolomic analysis (Fig. 3b) was also performed to identify signature metabolites for anaerobic hydrocarbon biodegradation[18,39].

**Diverse Euryarchaeota mediate anaerobic oxidation of methane and other short-chain alkanes.** Certain archaea activate short-chain alkanes (methane, ethane, propane, and butane) for anaerobic degradation using methyl/alkyl-coenzyme M reductases[17]. Sequences encoding the catalytic subunit of this enzyme (*mcrA*) were detected in metagenomes at all sediment depths except 0 cmbsf, with the highest abundance found at 60 cmbsf (Supplementary Fig. 14). A total of 20 MAGs within Euryarchaeota harbored *mcrA* sequences (Supplementary Data 10). Genome trees showed these microorganisms belonged to representatives of three families of hydrogenotrophic methanogens (*Methanomicrobiaceae*, *Methanosarcinaceae*, and *Methanosaetaceae*)[40], two clusters of anaerobic methanotrophs (ANME-1 and ANME-2)[41], and two lineages that have been shown to catalyze non-methane alkane oxidation (the GOM-Arc1 lineage, and a novel sister lineage to *Ca.* Syntrophoarchaeum)[17,27] (Fig. 4a). In agreement with this, phylogenetic analysis of *mcrA* sequences

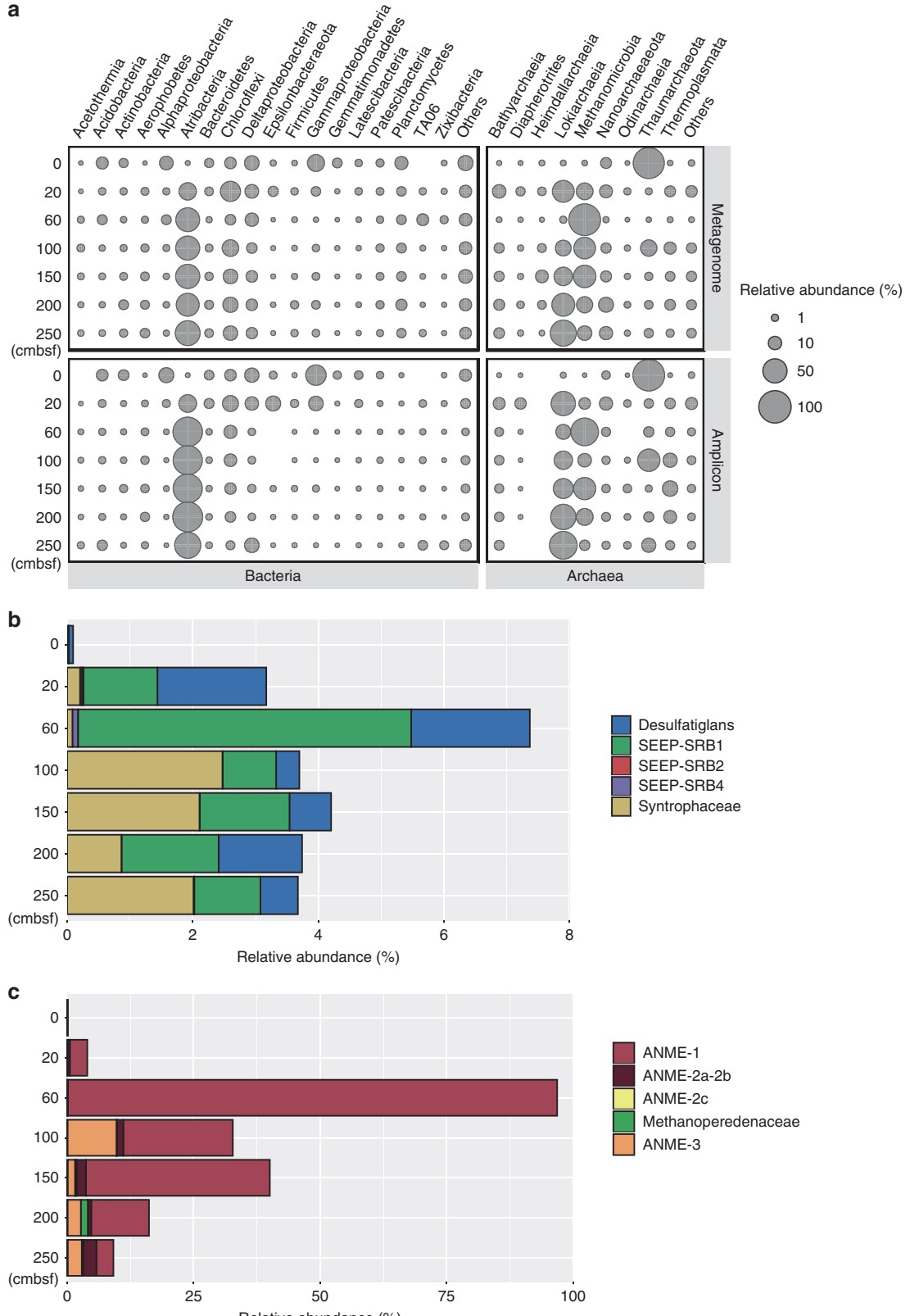

**Fig. 2 Compositions of microbial communities across sediment depths based on 16S rRNA genes. a** Relative abundances of bacterial and archaeal taxa at different sediment depths. Top panels: 16S rRNA gene fragments derived from metagenomic libraries using the phyloFlash pipeline. Bottom panels: 16S rRNA gene amplicons. Bacterial groups are shown on the left and archaeal groups are shown on the right. **b** Relative abundances of anaerobic bacterial hydrocarbon degraders among bacteria inferred from 16S rRNA gene fragments in metagenomes. **c** Relative abundances of anaerobic archaeal hydrocarbon degraders among archaea inferred from 16S rRNA gene fragments in metagenomes.

from these MAGs resolved three major groups (Fig. 4b), i.e., the canonical group clustering with methanogens and ANMEs, and two divergent groups clustering with GOM-Arc1 (e.g., *Ca.* Argoarchaeum)[17] and *Ca.* Syntrophoarchaeum[19].

Metabolic pathways involved in the oxidation of methane and non-methane gaseous alkanes were reconstructed[17] (Fig. 5). Twelve MAGs harbor canonical *mcrA* genes that cluster with ANME-1 and ANME-2 methanotrophs, along with with *fwd, ftr,*

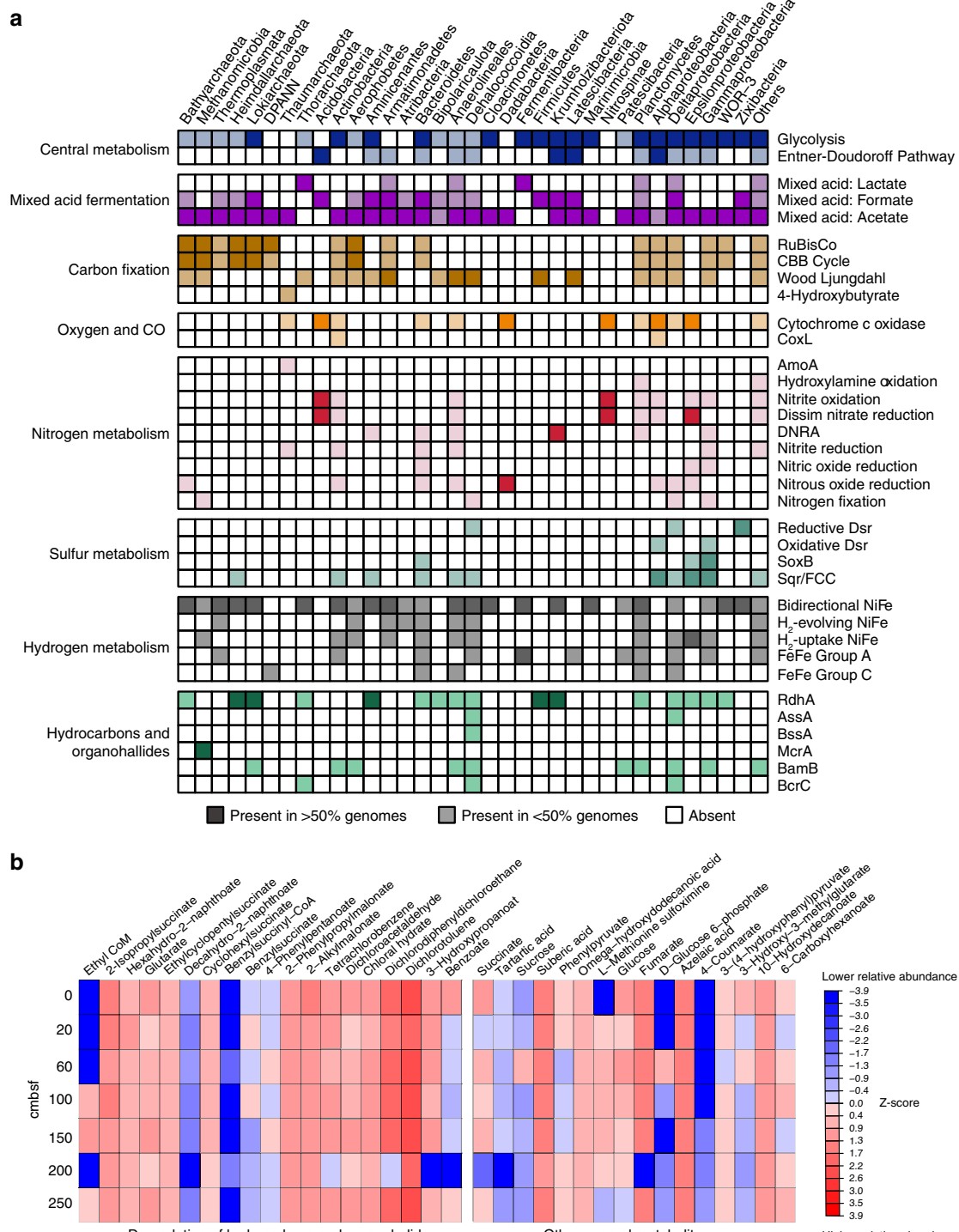

**Fig. 3 Functional profiles of different microbial groups and metabolites identified at different depths. a** Occurrence of core metabolic genes or pathways in phylogenetic clusters of MAGs. Phyla with only one representative MAG in the data set ($n = 10$) and MAGs that could not be classified ($n = 10$) are grouped together in the right-most column (Others). Dark and light color shading indicate gene presence in >50% and 1–50% of the genomes in each phylogenetic cluster, respectively. Complete lists of metabolic genes or pathways can be found in Supplementary Data 5. Detailed gene lists for each pathway indicated can be found at: https://github.com/bjtully/BioData/blob/master/KEGGDecoder/KOALA_definitions.txt. **b** Metabolites identified in sediment porewater from down core sediment subsamples. Metabolite levels were measured using HPLC Orbitrap mass spectrometry and are expressed as cumulative sum logarithmically normalized peak areas of triplicates. Alkyl-/arylalkylsuccinates and ethyl-CoM are signature metabolites for anaerobic hydrocarbon degradation, whereas other compounds shown can be derived from multiple biological or abiotic processes.

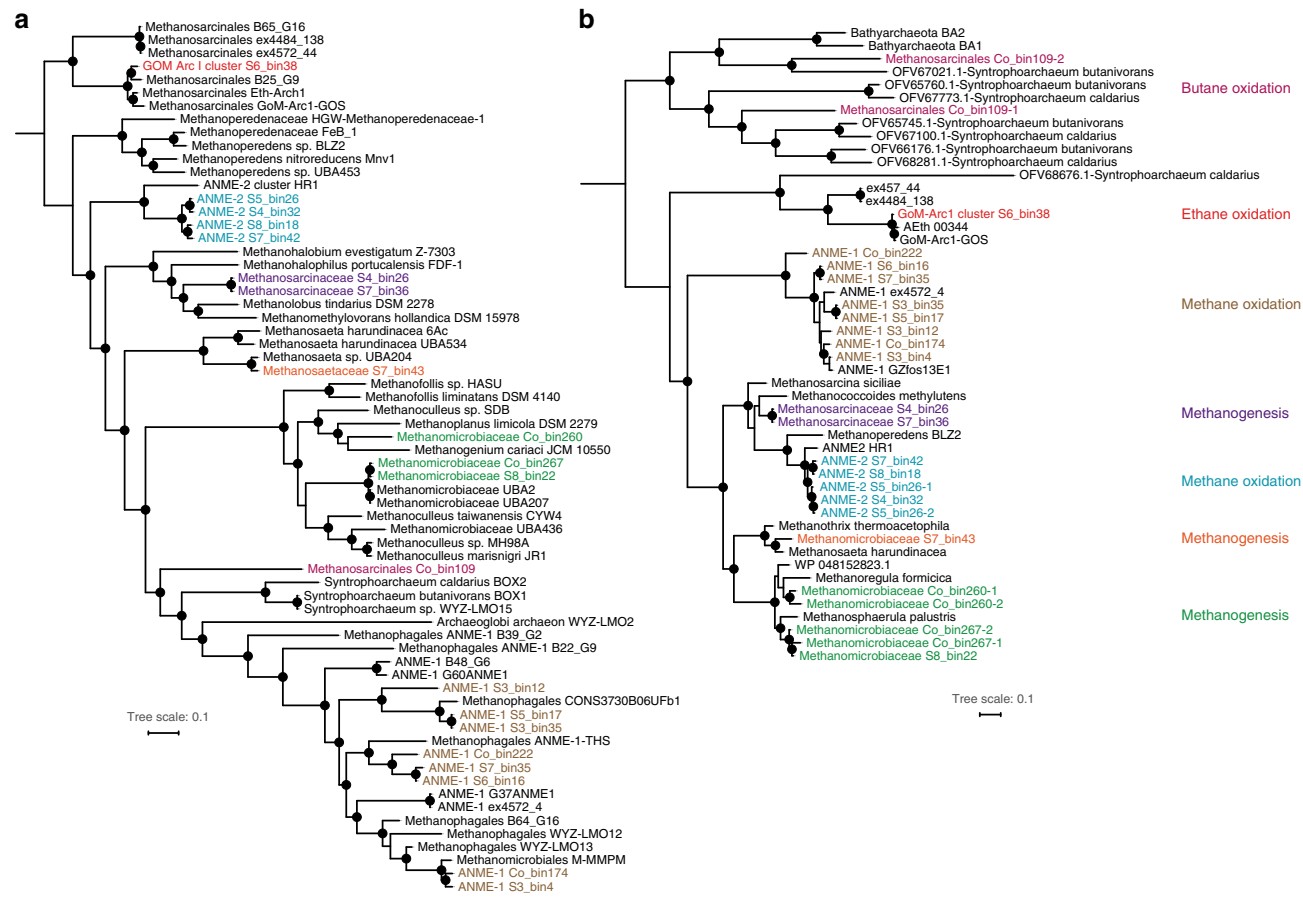

**Fig. 4 Maximum-likelihood phylogenetic trees of MAGs and the detected methyl-coenzyme M reductase (McrA) sequences. a** Phylogenomic tree constructed based on alignments of 43 conserved protein sequences from *Methanomicrobia* MAGs; **b** phylogenetic tree constructed based on alignments of amino-acid sequences of *mcrA* genes. Bootstrap values of >70% are indicated with black circles. MAGs and McrA sequences corresponding to the same alkane metabolisms are highlighted in the same color in the two trees. Scale bars correspond to percent average amino-acid substitution over the alignment, for both trees. Sequences for amino acids used to construct McrA trees can be found in Supplementary Data 15.

*mch, mtd, mer/metF/fae-hps* and *mtr* that mediate subsequent steps in the tetrahydromethanopterin-dependent "reverse methanogenesis" pathway[10,42] for oxidation of methyl-CoM to $CO_2$ (Fig. 5a and Supplementary Data 10). Some MAGs lack specific genes in this pathway, likely reflecting variability in genome completeness (61–96%). Based on genome and gene trees, S6_bin38 is closely related to putative ethane oxidizers within the GoM-Arc1 group[26,27,40], including the verified anaerobic ethane oxidizer *Ca.* Argoarchaeum ethanivorans Eth-Arch1[17] (Fig. 4). In agreement with this, the signature metabolite ethyl-coenzyme M (ethyl-CoM) was detected in most of the deeper sediments (Fig. 3b). Like other GoM-Arc1 genomes, S6_bin38 encodes methyltransferases that potentially transfer the thioether (ethyl-CoM) derived from ethane activation to a thioester (acetyl-CoA), as well as enzymes to mediate acetyl-CoA cleavage (acetyl-CoA decarbonylase/synthase) and stepwise dehydrogenation of the derived $C_1$ units (oxidative Wood-Ljungdahl pathway) (Fig. 5b and Supplementary Data 10). Like other members of the GoM-Arc1 group, S6_bin38 lacks the β-oxidation pathway which is unnecessary for anaerobic ethane oxidation[17].

*Methanosarcinales* Co_bin109 likely has potential to oxidize butane and fatty acids present in the sediments (Table 1 and Fig. 3b). This genome contains two *mcrA* genes that cluster with high bootstrap support to the divergent alkyl-CoM reductases from *Ca.* Syntrophoarchaeum (Fig. 4b) that is capable of anaerobic degradation of butane and possibly

propane[19]. Also encoded by this MAG are heterodisulfide reductase subunits (*hdrABC*) to reoxidize cofactors[28], methyltransferases that potentially convert butyl-thioether to the butyryl-thioester, and the β-oxidation pathway to enable complete oxidation of the butyryl-thioester (Fig. 5c and Supplementary Data 10). The presence of short-chain acyl-CoA dehydrogenase (*acd*), butyryl-CoA dehydrogenase (*bcd*), and long-chain acyl-CoA synthetase (*fadD*) may allow *Methanosarcinales* Co_bin109 to oxidize long-chain fatty acids; this is similar to the basal *Archaeoglobi* lineage *Ca.* Polytropus marinifundus, which encodes two divergent McrA related to those found in *Ca.* Bathyarchaeota and *Ca.* Syntrophoarchaeum[43]. Consistent with this, Co_bin109 encodes a Wood-Ljungdahl pathway (including carbon monoxide dehydrogenase/acetyl-CoA synthase complex) for complete fatty-acid oxidation (Supplementary Data 10).

**Members of Chloroflexi and *Deltaproteobacteria* potentially degrade liquid alkanes and aromatic hydrocarbons.** Other community members are predicted to degrade aromatic hydrocarbons and *n*-alkanes detected in the sediments (Table 1 and Supplementary Table 1) via hydrocarbon addition to fumarate[44]. Sequences encoding the catalytic subunits of alkylsuccinate synthase (*assA*), benzylsuccinate synthases (*bssA*), and naphthyl-methylsuccinate synthases (*nmsA*) were detected in metagenomes at all sediment depths except 0 cmbsf (Supplementary Fig. 14). Nine MAGs assigned to *Dehalococcoidia, Desulfobacterales*, and

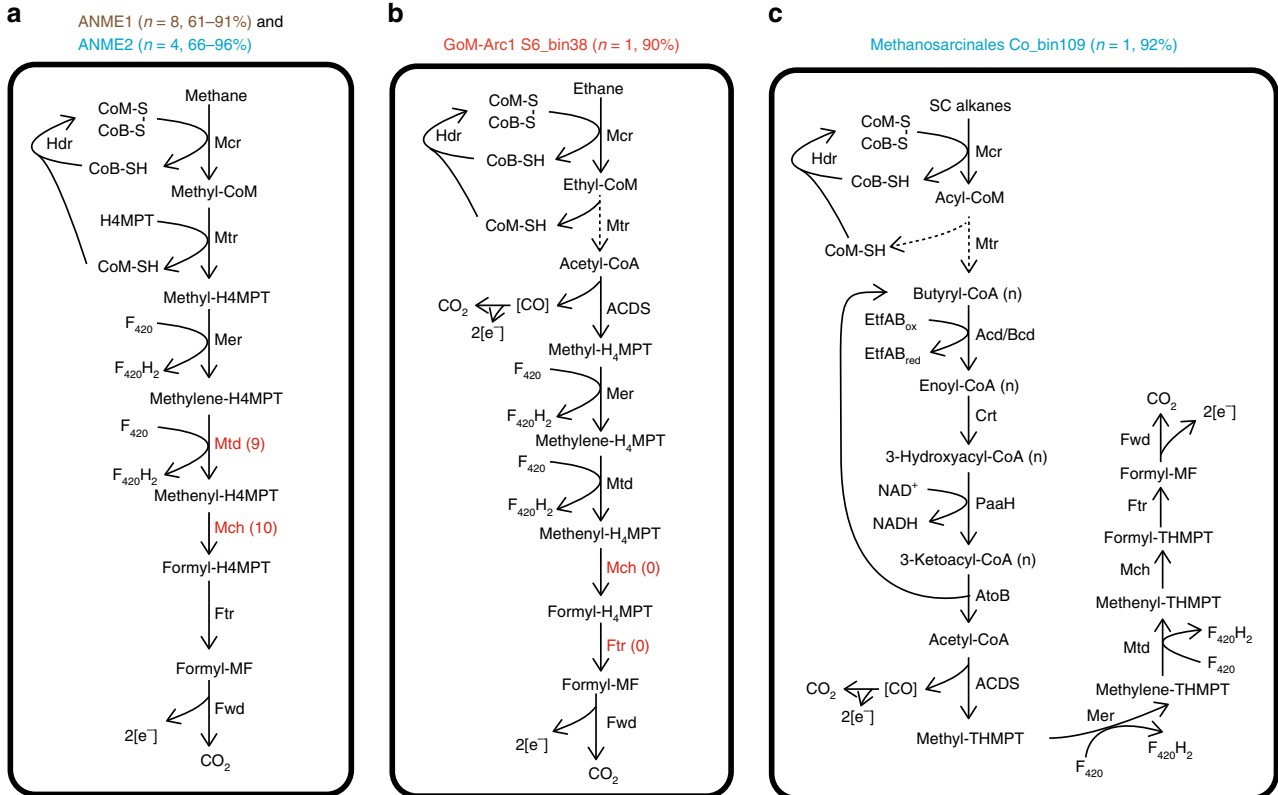

**Fig. 5 Predicted metabolic models for anaerobic oxidation of gaseous alkanes based on archaeal MAGs harboring *mcrA* genes. a** Archaeal oxidation of methane; **b** archaeal oxidation of ethane; and **c** archaeal oxidation of butane. Red font indicates that not all MAGs retrieved encode the enzyme (numbers of MAGs encoding the enzyme are indicated in parentheses). Dashed lines indicate steps catalyzed by unconfirmed enzymes. SC alkanes, short-chain alkanes. Percentages above each panel indicate the completeness of the corresponding MAGs estimated by CheckM. Details on the annotation of the enzymes are presented in Supplementary Data 10.

*Syntrophobacterales* encode alkylsuccinate synthase known to mediate *n*-alkane activation (Fig. 6a and Supplementary Data 11). Phylogenetic analysis confirms that these *assA* genes cluster closely with those of experimentally validated alkane oxidizers *Desulfatibacillum aliphaticivorans* DSM 15576 and *Desulfosarcina* sp. BuS5 within *Deltaproteobacteria*[8,45] (Supplementary Fig. 15). In most of these MAGs, more than one *assA* sequence variant was identified, suggesting some bacteria may activate multiple substrates by this mechanism[46,47]. In addition to *assA* sequences, *Dehalococcoidia* Co_bin57 and Co_bin289 also encode related glycyl-radical enzymes clustering with benzylsuccinate synthases and naphthylmethylsuccinate synthases (Supplementary Fig. 15 and Supplementary Data 11), known to mediate the anaerobic degradation of toluene or similar aromatic compounds. Metabolomic analysis detected four succinic acid conjugates involved in hydrocarbon activation, including conjugates of both toluene and propane (Figs. 3b, 6). Most of the MAGs encoding *assA* or *bssA* genes (except *Dehalococcoidia* Co_bin57) also encode the required genes to further process the alkyl-/arylalkylsuccinate compounds, convert them to acetyl-CoA through the *β*-oxidation pathway, and regenerate fumarate through the methylmalonyl-CoA pathway (Fig. 6a, c; Supplementary Data 11). Accordingly, succinate and fumarate were also detected in the sediments (Fig. 3b).

Anaerobic hydrocarbon degradation depends on subsequent oxidation of acetyl-CoA. This can be achieved through a single-cell process (e.g., coupled to sulfate respiration) or via syntrophic interaction with another cell (e.g., with methanogens)[8,45]. Like the related deltaproteobacterial isolate NaphS2, *Desulfobacterales* MAGs S4_bin49 and S6_bin7 contain genes for both the

Wood-Ljungdahl pathway and dissimilatory sulfate reduction, suggesting the same organism can couple alkane mineralization directly to sulfate reduction (Supplementary Data 11). By contrast, other MAGs containing genes for hydrocarbon addition to fumarate, including *Dehalococcoidia* and *Syntrophobacterales*, apparently lack terminal reductases or complete tricarboxylic acid cycles. These organisms may be obligate fermenters dependent on syntrophy with respect to the oxidation of *n*-alkanes, as further evidenced by identification of genes for mixed-acid fermentation and hydrogen production in these genomes (Supplementary Data 5 and 11) and as reported in closely related organisms from both lineages[46].

Detection of *bssA* and *nmsA* suggest that these cold seep microbial communities are also capable of utilizing aromatic hydrocarbons that were detected in the gas chromatography mass spectrometry analysis (Supplementary Table 1 and Supplementary Fig. 14). Metabolites produced after initial activation normally channel into the central benzoyl-CoA degradation pathway (Fig. 6b). Benzoyl-CoA, as a universal biomarker for anaerobic degradation of aromatic compounds[48], is reduced by benzoyl-CoA reductases of the ATP-dependent class I pathway (*bcr* genes; e.g., in *Thauera aromatica*) or ATP-independent class II pathway (*bam* genes; e.g., in sulfate reducers). These genes were detected in 13 bacterial MAGs, including *Dehalococcoidia*, *Anaerolineae*, *Deltaproteobacteria*, Planctomycetes, *Gammaproteobacteria*, and candidate phylum AABM5-125-24. Genes for further processing these compounds to 3-hydroxypimelyl-CoA (i.e., *oah*, *dch*, and *had*) and acetyl-CoA (*β*-oxidation pathway) were also detected (Fig. 6c and Supplementary Data 12). Various compounds related to the benzoyl-

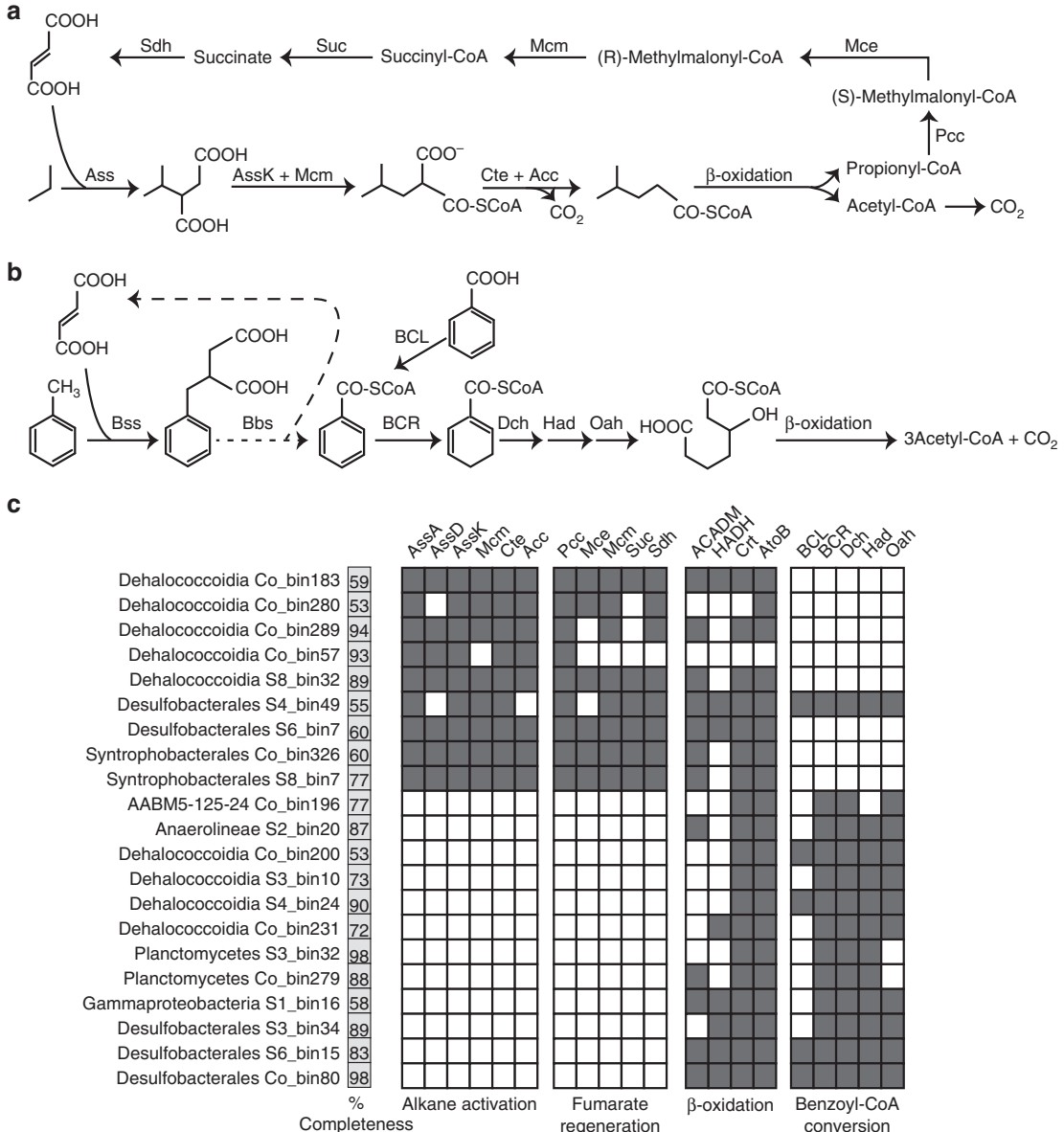

**Fig. 6 Predicted metabolic models for anaerobic degradation of alkanes and benzoate based on bacterial MAGs harboring *assA* and *bcr* genes.** Proposed metabolic pathways for anaerobic degradation of **a** propane and **b** toluene via addition to fumarate. Propane and toluene here serve as examples for illustrating anaerobic degradation of alkane and aromatic hydrocarbons more generally. **c** Occurrence of protein encoding genes involved in these pathways in bacterial MAGs, as a presence/absence (filled/white squares) matrix. Genome completeness (%) for each MAG as estimated by CheckM is indicated. Details on the annotation of the enzymes are presented in Supplementary Data 11 and 12.

CoA degradation pathway were also detected, including benzoate and glutarate (Fig. 3b).

**Depth distributions and in situ replication rates of hydrocarbon-oxidizing organisms.** To put the metabolic functions of anaerobic hydrocarbon degraders into ecological perspective, relative abundances and cell replication rates were assessed in different sediment depths. The 376 bacterial and archaeal genomes were dereplicated at the species level (i.e., 95% average nucleotide identity clustering) to avoid arbitrary mapping between representatives of highly similar genomes. This process yielded a total of 296 bacterial and archaeal species clusters (Supplementary Data 13), with the MAG of highest genome quality from each species cluster picked as a representative. Based on metabolic pathway reconstruction, the capacities for anaerobic degradation of hydrocarbons were detected in members from

Euryarchaeota, Chloroflexi and *Deltaproteobacteria* (Fig. 3a and Supplementary Data 5). Distributions of these organisms spanned from the near-surface sediment horizon to the deeper zones where sulfate was depleted, with overall relative abundance ranging from 1–42% at different sediment depths (Fig. 7 and Supplementary Data 13). MAGs corresponding to Atribacteria, Lokiarchaeota and Chloroflexi were also differentially enriched as a function of sediment depth (Supplementary Data 13), consistent with their dominance in these sediments (Fig. 2a) and their occurrence in the deep marine biosphere more generally[49,50].

MAGs with the highest relative abundance belonged to the ANME-1 lineage, and together made up >40% of the microbial community at 60 cmbsf (Fig. 7 and Supplementary Data 13), with three distinct ANME-1 species (S3_bin4, Co_bin174 and S3_bin12) being particularly abundant (27.47%, 7.33%, and 4.69%, respectively). The ANME-2 lineage on the other

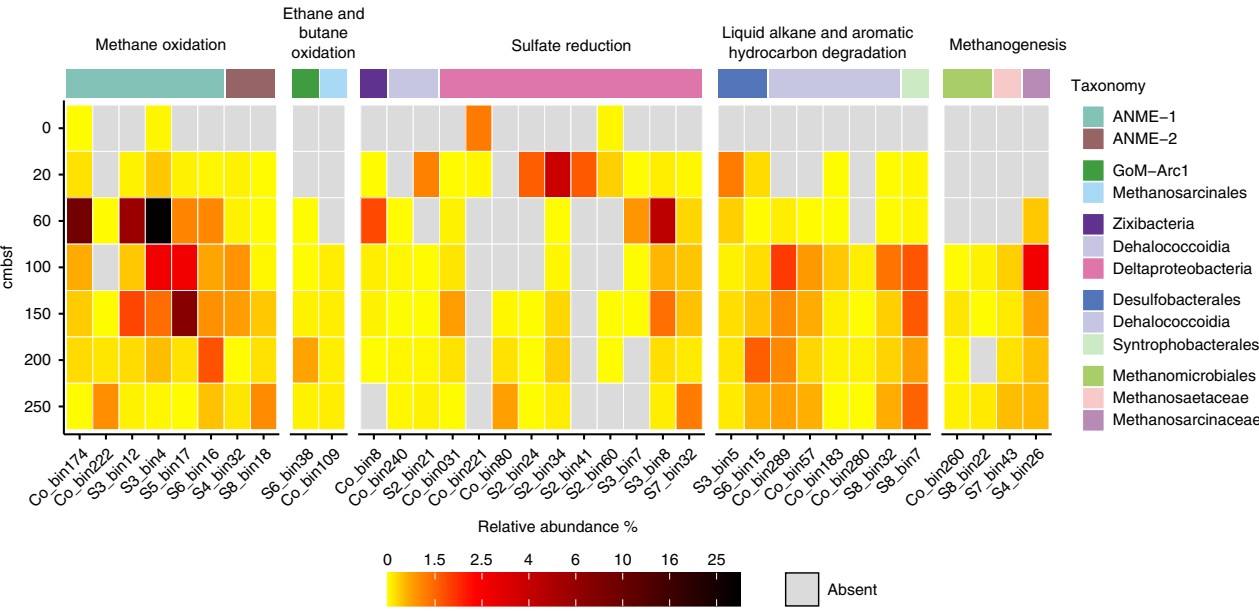

**Fig. 7 Depth distributions of anaerobic hydrocarbon oxidizers and their potential partners inferred from metabolic pathway reconstructions.** Species-level portrayal, with the MAG of highest genome quality from each species cluster being picked as the representative. Details on species clusters, relative abundances and replication rates of all microorganisms are presented in Supplementary Data 13.

hand was rare in this sediment, comprising only 0.04% of the community at 60 cmbsf. This observation is supported by 16S rRNA gene amplicon sequencing (Supplementary Data 1) and stands in contrast to other thermogenic hydrocarbon seeps, e.g., in the Gulf of Mexico, where sediments are dominated by ANME-2 lineages[10,11]. In agreement with analysis of 16S rRNA gene fragments in metagenomic libraries (Fig. 2c), ANME-1 were also present in deeper sulfate-depleted sediments, with cumulative relative abundances ranging from 0.57% to 8.18% (Fig. 7). Replication rates calculated using iRep[51] suggested that ANME-1 were active in these deeper sediments. For example, ANME-1 S3_bin4 had higher replication rates (iRep values up to 2.03) as compared to 1.20 at 60 cmbsf (Supplementary Data 13). Ethane and butane oxidizers (S6_bin38 and Co_bin109) were also mostly found in sulfate-depleted sediments, where they represented a minor fraction of the community (0.04–0.25%). The iRep value of S6_bin38 at 200 cmbsf was 1.37, suggesting that most cells were undergoing genome replication[51], which is consistent with the detection of ethyl-CoM in the same sediment, despite the absence of sulfate (Fig. 3b).

For anaerobic hydrocarbon oxidation to be energetically favorable, it must be coupled to reduction of an electron acceptor either directly or through interspecies electron transfer to a syntrophic partner. None of the canonical terminal reductases (e.g., for iron, sulfate, and nitrate reduction) reported in other studies[28,40] were detected in MAGs representing alkane-oxidizing archaea (Supplementary Data 5). Various sulfate reducers from Deltaproteobacteria and Zixibacteria were detected with full gene sets for dissimilatory reduction of sulfate to sulfide (sat, aprAB, and dsrAB) (Supplementary Fig. 5 and Supplementary Data 5). Of these, SEEP-SRB1 S3_bin8 peaked (3.4% of the community) at 60 cmbsf, which likely represented a sulfate-methane transition zone. This coincides with the highest relative abundance of ANME-1 (Figs. 2b and 7), in agreement with observations that these two groups operate together as syntrophic partners[12]. The presence of alkane-oxidizing archaea in sulfate-depleted sediments suggests that alkane oxidation might be linked to the reduction of alternative electron acceptors. Thirty bacterial members belonging to lineages such as Epsilonproteobacteria and Gammaproteobacteria were predicted to use nitrate or nitrite

as electron acceptors, but only dominated at the upper sediment layers, similar to the observation in gene-centric analyses (Supplementary Fig. 14 and Supplementary Data 5). These genomic analyses seem to rule out nitrate reducers as possible syntrophic partners. Even though no genomes were detected to encode genes for iron/manganese reduction (i.e., mtrA and mtrC), multiheme c-type cytochromes, hypothesized to mediate electron transport to syntrophic bacteria and directly to iron oxides[16], were found in these archaeal MAGs (Supplementary Data 14). Therefore, metal reduction may be coupled with oxidation of short-chain alkanes in deeper sediments. Other unknown process might also possibly consume electrons produced from oxidation of short-chain alkanes. For example, it was recently proposed that bacterium-independent anaerobic methane oxidation could be coupled to reduction of elemental sulfur and polysulfide[10]. Consistent with this model, genes encoding sulfide:quinone oxidoreductase-like proteins were detected, e.g., in ANME-1 S3_bin4 and Co_bin174.

Bacterial liquid alkane and aromatic hydrocarbon degraders were observed in higher relative abundance mainly in sulfate-depleted sediments (1.59–2.69% in 100–250 cmbsf vs 0–0.60% in 0–60 cmbsf sediments; Fig. 7 and Supplementary Data 13), consistent with most MAGs lacking genes for respiration (Supplementary Data 5). Members of the Syntrophobacterales, such as Smithella and Syntrophus, have been shown to be able to degrade alkanes via addition to fumarate under methanogenic conditions in syntrophic association with methanogens[8,52]. Several MAGs from Methanomicrobiales, Methanosaetaceae, and Methanosarcinaceae found at the same depths were identified as typical methanogens (Figs. 4a and 7), thus syntrophic alkane mineralization via hydrogenotrophic or acetoclastic methanogenesis represents another plausible route for hydrocarbon biodegradation in this setting (Supplementary Data 5). Accordingly, potential alkane degraders like Syntrophobacterales S8_bin7 were mainly found in the sulfate-depleted sediments, co-occurring with methanogens, e.g., Methanomicrobiaceae S4_bin26 (Fig. 7). Similarly, Dehalococcoidia Co_bin289 and Co_bin57 showed high replication rates (1.3–2.1) in sulfate-depleted sediments, suggesting a syntrophic coupling of long-chain alkane degradation by Chloroflexi with methanogenesis (Supplementary Data 13).

## Discussion

Biodegradation by sedimentary microbial communities is an important mechanism that controls natural emissions of hydrocarbons from the deep subsurface. This study combines geophysics, geochemistry, metagenomics, and metabolomics to characterize a newly discovered deep sea cold seep in unprecedented detail. Located in the poorly explored Scotian Basin in the NW Atlantic, this seep is associated with salt tectonics, similar to other oil- and gas-influenced seabed habitats[8,13]. Unlike hypersaline brine seeps like those in the Gulf of Mexico[13,53], this Scotian Basin sediment has a similar salinity to seawater in the cold deep sea. Through gene- and genome-centric analyses, diverse novel uncultured and undescribed anaerobic bacteria (e.g., new *Dehalococcoidia* lineages) and archaea (e.g., a novel sister lineage to *Ca.* Syntrophoarchaeum) were inferred to consume upward migrating gaseous and liquid thermogenic hydrocarbons in situ. Metagenomic and metabolomic evidence reveal that hydrocarbons are degraded through alkyl-CoM reductase and fumarate addition pathways. Identification of succinate derivatives[39] (e.g., 2-isopropylsuccinate and benzylsuccinate) and ethyl-CoM in porewater, together with determination of carbon stable isotopic signatures for hydrocarbon gases (e.g., ethane and propane) and carbon dioxide[32] provide direct evidence of active biodegradation in these deep sea sediments. These findings point to geological phenomena selecting for bacteria and archaea with differing and complementary mechanisms for metabolizing hydrocarbon substrates, with different bacteria oxidizing liquid alkanes and aromatic hydrocarbons and different archaea oxidizing methane and other short-chain alkane gases.

This cultivation-independent genomic investigation of in situ seabed biogeochemistry points to a broader role for archaeal multi-carbon alkane oxidation than previously suggested. This recently described process[17] co-occurs with anaerobic methane oxidation by ANME archaea in these cold seep sediments. Anaerobic methane oxidizers are most abundant, which is unsurprising given high concentrations of methane relative to other short alkanes; however, investigating an environment that still has relatively high concentrations of $C_{2+}$ gases (up to 6.5%), the potential and importance of both GoM-Arc1- and *Syntrophoarchaeum*-like archaea degrading thermogenic hydrocarbons in deep seabed sediments is shown for the first time. Previous observations of these phenomena and enrichment of the corresponding microorganisms have focused mainly on hydrothermally heated sediments[15,27,40] and hypersaline methane seeps[53,54]. The results presented here therefore point to a widespread significance for these microbial groups in submarine carbon cycling throughout the global ocean, regardless of salinity and temperature.

Combined metagenomics and metabolomics also suggest that a range of electron acceptors support hydrocarbon degradation in cold seeps. Depth profiles showed that bacteria capable of degrading liquid alkane and aromatic hydrocarbons were mainly detected in sulfate-depleted sediments, suggesting that they mediated hydrocarbon metabolism as part of methanogenic alkane-degrading consortia[8,44,45]. Also as expected, ANME-1 were found to be most abundant at the apparent sulfate-methane transition zone, in agreement with the well-documented syntrophy between methane-oxidizing archaea and sulfate-reducing bacteria in such settings[22]. These findings support that the cooperation of key bacterial and archaea hydrocarbon degraders with their partners is important for hydrocarbon degradation at deep sea cold seep sediments. More surprisingly, contrary to the paradigm that gaseous alkane-oxidizing archaea normally occur where alkanes and sulfate coexist, we observed that ANME-1, ethane-oxidizing and butane-oxidizing archaea were also actively present in sulfate-depleted sediments. Based on their occurrence

and activity in shallow coastal sediment cores from Aarhus Bay and White Oak River Estuary, ANME-1 has been proposed to be capable of reverse methanogenesis in sulfate-methane transition or sulfate-depleted sediments[55–57]. It is also possible that members of ANME-1 are contributing to methane production in the deep sea cold seep sediments investigated here. Given high concentrations of hydrocarbon gases and associated signature metabolites (e.g., ethyl-CoM), another scenario is that these alkane-oxidizing archaea utilize electron acceptors other than sulfate, e.g., iron oxides, humic redox shuttles, elemental sulfur and polysulfide[10,58]. Thus, despite sulfate availability being thought to limit gaseous alkane oxidation, other electron acceptors may substitute. This advances our understanding of the capacity and mechanisms for anaerobic methane, ethane, propane, and butane oxidation in archaea[17,19,40,58].

## Methods

**Sampling and geochemical characterization**. This study provides a detailed analysis of a 3.44-meter-long piston core taken from the Scotian Basin seabed (43.010478 N, 60.211777 W) in 2306 m water depth on the Scotian Slope. The coring location was chosen based on seismic interpretation using Petrel that focused on identifying amplitude anomalies for direct hydrocarbon indicators. One such location at this location was inferred to be associated with a possible seabed seep by interpreting a subsurface hydrocarbon migration pathway in the form of a fault to surface. Following piston coring, sediment subsamples were collected immediately from the base of the core and stored in gas-tight isojars flushed with $N_2$ for headspace gas analysis. Multiple depths ranging from the deepest portion to within approximately one meter of the top of the core, were subsampled for geochemical analysis. Additional intervals were preserved separately for microbiological analyses. Detailed subsampling depths can be found in Tables 1 and 2 as well as Supplementary Table 1.

Hydrocarbon compositions of headspace gas samples were analyzed using an Agilent 7890 A RGA gas chromatograph equipped with Molsieve and Poraplot Q columns and a flame ionization detector. Stable carbon and hydrogen isotopic signatures were determined by Trace GC2000 equipped with a Poraplot Q column, connected to a Thermo Finnigan Delta plus XP isotope ratio mass spectrometer (IRMS). Sediment samples were analyzed for TOC and EOM using an Agilent 7890 A RGA gas chromatograph equipped with a CP-Sil-5 CB-MS column. A Micromass ProSpec-Q instrument was used for determination of saturated and aromatic fractions. Stable carbon isotope analyses of these fractions were determined on a Eurovector EA3028 connected to a Nu Horizon IRMS. Experimental procedures on these measurements followed "The Norwegian Industry Guide to Organic Geochemical Analyses, Edition 4.0 (30 May 2000)".

**Sulfate measurement**. Porewater sulfate concentrations were measured in a Dionex ICS-5000 reagent-free ion chromatography system (Thermo Scientific, CA, USA) equipped with an anion-exchange column (Dionex IonPac AS22; $4 \times 250$ mm; Thermo Scientific), an EGC-500 $K_2CO_3$ eluent generator cartridge and a conductivity detector. Measured values of sulfate concentrations were corrected with method-related factors based on a series of standard samples.

**Metabolomic analysis**. Porewater metabolites were extracted from sediment samples according to previously reported methods[4]. Mass spectrometric (MS) analysis was carried out using a Thermo Scientific Q-Exactive HF Hybrid Quadrupole-Orbitrap mass spectrometer with an electrospray ionization source coupled to ultra high-performance liquid chromatography. Data were acquired in negative ion mode using full scan from 50–750 $m/z$ at 240,000 resolution with an automatic gain control (AGC) target of 3e[6] and a maximum injection time of 200 ms. For MS/MS fragmentation, an isolation window of 1 $m/z$ and an AGC target of 1e[6] was used with a maximum injection time of 100 ms. Data were analyzed for specific $m/z$ ratios using MAVEN software[59].

**16S rRNA gene amplicon sequencing**. DNA was extracted from sediment samples using the PowerSoil DNA Isolation Kit (12888-50, QIAGEN). Amplification of the v3-4 region of bacterial 16S rRNA genes and the v4-5 region of archaeal 16S rRNA genes, used primer pairs SD-Bact-0341-bS17/SD-Bact-0785-aA21 and SD-Arch-0519-aS15/SD-Arch-0911-aA20, respectively[60]. Amplicon sequencing was performed on a MiSeq benchtop sequencer (Illumina Inc.) using the $2 \times 300$ bp MiSeq Reagent Kit v3. Reads were quality controlled and then clustered into operational taxonomic units (OTUs) of >97% sequence identity with MetaAmp[61]. Taxonomy was assigned together with the SILVA database[62] (release 132).

**Quantitative PCR**. Quantitative polymerase chain reaction (qPCR) analyses were performed on the new DNA extracts using PowerSoil DNA Isolation Kit (12888-

50, QIAGEN) to estimate the abundance of bacteria and archaea at different depths in the sediment core. The mass of sediment used for DNA extraction was typically 0.5 g and was always recorded. PCR reactions were set up using Bio-Rad SsoAdvanced Universal SYBR Green Supermix. Amplification of bacterial and archaeal 16S rRNA genes used domain-specific primers B27F-B357R and A806F-A958R, respectively. Triplicate PCR was performed on a Thermo Scientific PikoReal Real-Time PCR Instrument, using reaction conditions described previously[63]. Results were recorded and analyzed by PikoReal software 2.2.

**Metagenome sequencing**. DNA was extracted from the sediment samples using the larger format PowerMax Soil DNA Isolation Kit (12988-10, QIAGEN) according to the manufacturer's instructions. Metagenomic library preparation and DNA sequencing using NextSeq 500 System (Illumina Inc.) were conducted at the Center for Health Genomics and Informatics in the Cumming School of Medicine, University of Calgary.

**Microbial diversity analysis**. SingleM (https://github.com/wwood/singlem) was applied to raw metagenome reads from each sample[34]. Shannon diversity was calculated based on SingleM counts on 14 single-copy marker genes. The vegan package was then used to calculate diversity based on the rarefied SingleM OTU table across each of the 14 marker genes. The average was taken as the Shannon index determination for each sample. To explore microbial composition of each sample, 16S rRNA gene fragments were recovered from metagenomic raw reads using the phyloFlash pipeline[35] together with SILVA database[62] (release 132).

**Assembly and binning**. Raw reads were quality-controlled by (1) clipping off primers and adapters and (2) filtering out artifacts and low-quality reads using the BBDuk function of BBTools (https://sourceforge.net/projects/bbmap/). Filtered reads were co-assembled using MEGAHIT[64] and were individually assembled using metaSPAdes[65]. For co-assembly, one additional metagenome (315 cmbsf) sequenced using the same method was also included. This depth was discarded for other analyses owing to suspicions that it was contaminated with seawater. Short contigs (<1000 bp) were removed from assemblies. For each assembly, binning used the Binning module within metaWRAP[37] (–maxbin2 –metabat1 –metabat2 options). Resulting bins were then consolidated into a final bin set with meta-WRAP's Bin_refinement module (-c 50 -x 10 options). All binning results were combined and dereplicated using dRep[66] (-comp 50 -con 10 options) at 99% average nucleotide identity clustering (strain level). After dereplication, a total of 376 dereplicated MAGs were obtained. SingleM v0.12.1 (https://github.com/wwood/singlem) was used to determine genome recovery efforts at genus level (singlem appraise –imperfect –sequence_identity 0.89).

**Calculating relative abundances and replication rates**. For producing indexed and sorted BAM files, quality-controlled reads from each sample were mapped to the set of dereplicated genomes at 95% average nucleotide identity clustering using BamM v1.7.3 "make" (https://github.com/Ecogenomics/BamM). To calculate relative abundance of each MAG within a microbial community at a given sediment depth, CoverM v0.4.0 "genome" (https://github.com/wwood/CoverM) was used to obtain relative abundance of each genome (parameters:–min-read-percent-identity 0.95 –min-read-aligned-percent 0.75 –trim-min 0.10 –trim-max 0.90).

Microbial replication rates were estimated with iRep[51] for high-quality dereplicated MAGs at 95% average nucleotide identity clustering (≥ 75% complete, ≤ 175 fragments/Mbp sequence, and ≤ 2% contamination). Replication rates were retained only if they passed the default thresholds: min cov. = 5, min wins. = 0.98, min $r^2$ = 0.9, GC correction min $r^2$ = 0.0. The require ordered SAM files were generated using the Bowtie2 (-reorder flag)[67].

**Functional annotations**. To compare abundances of metabolic genes at different sediment depths, all quality-controlled reads were aligned against comprehensive custom databases[68] using DIAMOND BLASTx[69] (cutoffs: e value: 1e-10, identity: 70%; best hits reserved).

For contigs, gene calling was performed using Prodigal (-p meta)[70]. Proteins were predicted against the KEGG database using GhostKOALA[71] and against the Pfam and TIGRfam HMM models using MetaErg[72]. For individual MAGs, completeness of various metabolic pathways was determined using KEGG Decoder[73] and KEGG-Expander (https://github.com/bjtully/BioData/tree/master/KEGGDecoder). Annotations of key metabolic genes were also confirmed by phylogenetic analyses as described below.

Genes involved in anaerobic hydrocarbon degradation were screened using BLASTp (cutoffs: e value 1e-20 + pident 30% + qcovs 70%) against local protein databases[4]. After removal of short sequences, genes were further manually curated using BLASTp against NCBI-based nr protein sequences by checking top hits to relevant genes. For identification of McrA and DsrA, protein sequences were screened against local protein databases[74] using BLASTp (cutoffs: e value 1e-20 + pident 30% + qcovs 70%). McrA and DsrA protein sequences were cross-checked against MetaErg annotations and phylogenetic analyses, whereas hydrogenases were confirmed and classified using the HydDB tool[75]. The dbCAN2 web server[76] was used for carbohydrate-active gene identification based on retaining proteins found by at least two of the tree tools (HMMER + DIAMOND + Hotpep) for

further analysis. To identify cytochrome C, MAGs with identified McrA were screened for proteins with at least one CXXCH motif[43]. These proteins were identified as cytochrome C if they also matched a protein domain related to cytochrome C using Batch web CD-search tool[77]. Psortb[78] was used to predict subcellular localizations.

**Taxonomic assignments of MAGs**. Taxonomic assessment of MAGs was initially performed by identification of 16S rRNA genes using anvi'o[79]. Predicted sequences (52 out of 376 MAGs) were aligned and classified using SILVA ACT[80]. Subsequently the taxonomy of each MAG was temporally assigned using GTDB-Tk[81] (using GTDB R04-RS89). Phylogenetic trees were reconstructed based on concatenation of 43 conserved single-copy genes using RAxML[82] settings: raxmlHPC-HYBRID -f a -n result -s input -c 25 -N 100 -p 12345 -m PROTCATLG -x 12345, as reported previously[4]. Bacterial and archaeal reference genomes were downloaded from NCBI GenBank. Finally, MAGs were identified to appropriate taxonomic levels according to the NCBI Taxonomy database, taking results of all three of the above methods into account.

**Phylogenetic analysis of metabolic genes**. For *mcrA*, gene sequences were aligned using the MUSCLE algorithm[83] (-maxiters 16) and trimmed using TrimAL[84] with parameters: –automated1. A maximum-likelihood phylogenetic tree was built using IQ-Tree[85], parameters: -st AA -m LG + C60 + F + G -bb 1000 -alrt 1000 -nt 20. For other key metabolic genes, sequences were aligned using the ClustalW algorithm included in MEGA7[86]. All alignments were manually inspected. For amino-acid sequences of the group 3 [NiFe]-hydrogenase large subunit, a neighbor-joining tree was constructed using the Poisson model with gaps treated with pairwise deletion, bootstrapped with 50 replicates and midpoint-rooted. For amino-acid sequences of other genes, maximum-likelihood trees of were constructed using the JTT matrix-based model with all sites, bootstrapped with 50 replicates and midpoint-rooted.

**Reporting summary**. Further information on research design is available in the Nature Research Reporting Summary linked to this article.

## Data availability

The links to the databases used in this study are listed below: Silva database (release 132): https://www.arb-silva.de/documentation/release-132/; NCBI Taxonomy database: https://www.ncbi.nlm.nih.gov/taxonomy; Pfam: https://pfam.xfam.org/; TIGRfam: https://tigrfams.jcvi.org/cgi-bin/index.cgi; KEGG GENES Database: https://www.genome.jp/kegg/genes.html; the custom database for anaerobic hydrocarbon degradation: https://www.nature.com/articles/s41467-019-09747-0#additional-information. DNA sequences have been deposited in NCBI BioProject databases under the accession number PRJNA598277. Individual assembly for MAGs can be found at figshare (https://figshare.com/s/bee9fd40f45054e71e8b). The authors declare that all other data supporting the findings of this study are available within the article and its supplementary information files, or from the corresponding authors upon request.

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

## Acknowledgements

This work was supported by Genome Canada, Genome Atlantic, and Genome Alberta through a Genomic Applications Partnership Program (GAPP) award. Ship time funding and support was provided by the Nova Scotia Department of Energy and Mines, and Natural Resources Canada. C.R.J.H. was awarded research funds for this work from Genome Canada (GAPP), a Canada Foundation for Innovation award (CFI-JELF 33752) and a Campus Alberta Innovates Program Chair. X.D. is supported by National Natural Science Foundation of China (grant no. 41906076) and the Fundamental Research Funds for the Central Universities (grant no. 19lgpy90). Metabolomics data were obtained at the Calgary Metabolomics Research Facility (CMRF), which is supported by the International Microbiome Center and the Canada Foundation for Innovation (CFI-JELF 34986) awards to I.A.L., who is supported by an Alberta Innovates Translational Health Chair. C.G. is supported by an ARC DECRA Fellowship (DE170100310) and an ARC Discovery Project (DP180101762). We thank Weiling Pi, Chuwen Zhang, Zexin Li, and Haoyu Lan for help with figure preparation, Ben Woodcroft for help with CoverM software, and Rhonda Clark for research support.

## Author contributions

C.R.J.H. obtained the funding for this project. X.D. and C.R.J.H. designed the study. X.D. processed metagenome data. J.E.R., S.M., R.A.G., and I.A.L. performed metabolomics analyses and data interpretation. D.C.C. was the chief scientist aboard the CCGS *Hudson* and was responsible for sediment sampling. M.F., J.W., and A.M. designed and performed petroleum geochemical analyses. N.M.M., D.C.C., and A.M. interpreted geophysical data. C.G. performed phylogenetic analysis of key metabolic genes. C.L. conducted amplicon sequencing and analyses. A.C. performed microbial diversity analyses. O.A. performed porewater sulfate measurements. S.W. and D.M. conducted qPCR analyses. X.D., C.G., and C.R.J.H. wrote the manuscript. All authors reviewed the results and participated in the writing of the manuscript.

## Competing interests

The authors declare no conflict of interest.
