## [Peer Review File · Nature Communications]

Reviewers' Comments:

Reviewer #1:

Remarks to the Author:

This study combined geochemical, metabolic, amplicon- and genomic characterization of a hydrocarbon-rich seep sediment core and its microbial populations, in order to show how different populations and pathways interact in the degradation of alkanes and aromatic compounds. The study was performed thoroughly and competently and is highly informative, but some "missing link" makes it hard for the different pieces to come together and to form a compelling story that befits this journal. However, this reviewer thinks the problem can be fixed.

A depth profile of sulfate-reducing bacteria, including potential methane- and alkane-degrading syntrophs (SEEP-SRB1, SEEP-SRB2, Desulfatiglans lineage, Smithella spp.), is essential for elucidating the changing pathways of hydrocarbon oxidation in this hydrocarbon-rich core, and it would support the discussion (for example line 444 ff.). It should be possible to see a transition from diverse hydrocarbon-oxidizing SRBs in the sulfate-reducing zone, towards methane oxidizers at the methane/sulfate interface, and finally non-sulfate-reducing, syntrophic members in methanogenic alkane-degrading consortia. This depth profile should be complemented by a matching depth profile of their hydrocarbon-degrading archaeal partners, ANMEs, Syntrophoarchaea, GoM-Arc1 etc. The intention to show this transition among different hydrocarbon-degrading populations is announced in the abstract [lines 33ff.], and some of the relevant populations are briefly highlighted in the introduction [lines 70-80].

The Results section does contain the relevant information in some form or another, but the reader is looking in vain for a clear summary on changing hydrocarbon degradation pathways and microbial groups in the sediment column: presumably aerobic or nitrate-reducing populations are found at the top, sulfate-reducing hydrocarbon degraders in the sulfate-rich zone, ANME consortia at the SMTZ, Syntrophoarchaeum and Ethanoperedens where alkanes and sulfate coexist, and finally methanogenic alkane-degrading consortia when sulfate runs out for good. The text is suggesting this message but it is actually not explicitly shown, at least not in the present version of the manuscript.

The key point of this study – its claim to fame - is to show how different microbial communities or consortia ["the redox-stratified seafloor microbiome" of the title] are taking turns in degrading and oxidizing hydrocarbons, depending on sediment depth and electron acceptor availability. Relevant pieces of information are distributed throughout the text or implied in various figures [for example, depth profiles of phylum-level lineages (Fig. 1), key functional genes (Fig. 2a) and key metabolites (Fig. 5) are shown], but there is no summary or synthesis figure to systematically visualize this key message.

It should be possible to look at the data afresh and to extract the information on hydrocarbon-degrading organisms and pathways, and their stratification in geochemical context, to design a strong figure that will summarize and wrap up the manuscript. Instead, the discussion ends with a mention of microbial necromass degradation [line 491]; yes, it exists, but wasn't this manuscript supposed to be about something else? Highlighting and actually showing the central message needs more attention.

Line 387: "both of which are known"

Line 432: "Anaerobic methane oxidizers..."

Andreas Teske

Reviewer #2:

Remarks to the Author:

The conclusion that "upward migrated thermogenic hydrocarbons . . . sustain diverse microbial communities" across a redox spectrum is not really a novel conclusion. Most methane seep sediment studies look at changes in the microbial communities in the sulfate rich and sulfate-depleted depths. Many of these studies have taken place in locations with a significant amount of input from thermogenic hydrocarbon, such as Guaymas Basin and the Gulf of Mexico. I think it is valuable to have another such study, especially in a new location. So, I support the publication of this work. But, it doesn't strike me as a very novel approach. Perhaps a more compelling rationale would be emphasizing what makes this location different than the others that have been examined.

As far as I can tell, downcore profiles were not made of methane, only of sulfate. The authors should make it clear that they are assuming the presence of a sulfate-methane transition zone, based on the depth where sulfate is depleted. Of course, there is methane in the bottom of the core, so it's possible there was a sulfate-methane transition zone. However, if this sample was in an active methane seep, which wouldn't be apparent with a gravity core, but is a clear possibility, then methane would be present at all layers, and there would be no sulfate-methane transition zone at all. So, the authors cannot say for sure that they have a sulfate-methane transition zone with no downcore methane profile.

The ANME-1 and GoM Arc 1 dominance seems similar to that of Lloyd et al., 2006 AEM. Perhaps the conclusions of that study should be compared to those of the current study.

It would be helpful to have a more thorough discussion of whether (and why) the metabolites are considered to be metabolic intermediates, metabolic products, or products of the thermal cracking.

Reviewer #1 (Remarks to the Author):

This study combined geochemical, metabolic, amplicon- and genomic characterization of a hydrocarbon-rich seep sediment core and its microbial populations, in order to show how different populations and pathways interact in the degradation of alkanes and aromatic compounds. The study was performed thoroughly and competently and is highly informative, but some “missing link” makes it hard for the different pieces to come together and to form a compelling story that befits this journal. However, this reviewer thinks the problem can be fixed.

Response: We appreciate this careful review and feedback. We have revised the manuscript in line with all of this reviewer’s comments, and with particular focus on depth-distributions of the seafloor microbiome in relation to its oxidation mechanisms for thermogenic hydrocarbons. Most notably we have substantially revised the Results and Discussion sections and added the new figure that the reviewer suggests. More detailed responses are provided below.

A depth profile of sulfate-reducing bacteria, including potential methane- and alkane-degrading syntrophs (SEEP-SRB1, SEEP-SRB2, Desulfatiglans lineage, Smithella spp.), is essential for elucidating the changing pathways of hydrocarbon oxidation in this hydrocarbon-rich core, and it would support the discussion (for example line 444 ff.). It should be possible to see a transition from diverse hydrocarbon-oxidizing SRBs in the sulfate-reducing zone, towards methane oxidizers at the methane/sulfate interface, and finally non-sulfate-reducing, syntrophic members in methanogenic alkane-degrading consortia. This depth profile should be complemented by a matching depth profile of their hydrocarbon-degrading archaeal partners, ANMEs, Syntrophoarchaea, GoM-Arc1 etc. The intention to show this transition among different hydrocarbon-degrading populations is announced in the abstract [lines 33ff.], and some of the relevant populations are briefly highlighted in the introduction [lines 70-80].

The Results section does contain the relevant information in some form or another, but the reader is looking in vain for a clear summary on changing hydrocarbon degradation pathways and microbial groups in the sediment column: presumably aerobic or nitrate-reducing populations are found at the top, sulfate-reducing hydrocarbon degraders in the sulfate-rich zone, ANME consortia at the SMTZ, Syntrophoarchaeum and Ethanoperedens where alkanes and sulfate coexist, and finally methanogenic alkane-degrading consortia when sulfate runs out for good. The text is suggesting this message but it is actually not explicitly shown, at least not in the present version of the manuscript.

Response: We have now added detailed information about the depth distributions of anaerobic hydrocarbon oxidizers and their partners in the Results section. To do so, we reanalyzed the metagenomic data, including (1) relative abundances of anaerobic hydrocarbon degraders inferred from 16S

rRNA gene fragments; (2) relative abundances and replication rates of species-level anaerobic hydrocarbon oxidizers inferred from metabolic reconstructions; (3) possible syntrophic partnerships, e.g. hydrocarbon-oxidizers with methanogens and sulfate reducers. The result of this re-analysis mostly agrees with the scenario that the reviewer proposes, and indeed makes our manuscript clearer.

The re-analysis also resulted in some interesting surprises; despite the well-known observation that gaseous alkane-oxidizing consortia normally occur where alkanes and sulfate coexist, ANME-1 were additionally found in sulfate-depleted sediments. Furthermore, ethane and butane oxidizers were predominantly detected in sulfate-depleted sediments. Activity by these groups at these depths is supported by cell replication rates and metabolomic analysis. These new observations are explained in detail in the Discussion, increase the novelty of this work, and set the stage for additional studies that interrogate the biogeochemical relationships and options for sedimentary oxidation of gaseous alkanes.

Changes made:

Revised Figures 2b, 2c, 3b and 7

L172-175: “ANME-1 (*Methanomicrobia*) comprised 97% of archaea at 60 cmbsf, consistent with the presence of its previously-observed syntrophic partner SEEP-SRB1 bacteria¹², suggesting that this depth was part of sulfate methane transition zone (Figures 2b and 2c).”

L190-196: “Several MAGs were affiliated with the Class *Methanomicrobia* (n = 24) within the phylum Euryarchaeota, including 12 MAGs belonging to ANME-1 and ANME-2 lineages. Bacterial MAGs were mostly represented by Chloroflexi (n = 93), Planctomycetes (n = 32), and *Deltaproteobacteria* (n = 29). Overall, the 376 MAGs captured the prevalent bacterial and archaeal lineages revealed by 16S rRNA gene analysis, representing 63.3-90.6% of the genera present in metagenomes for the deeper 20-250 cmbsf (cf. only 12.2% for 0 cmbsf).”

New Results sub-section L321-403 - Depth distributions and *in situ* replication rates of hydrocarbon-oxidizing organisms: “To put the metabolic functions of anaerobic hydrocarbon degraders into ecological perspective, relative abundances and cell replication rates were assessed in different sediment depths. The 376 bacterial and archaeal genomes were dereplicated at the species level (i.e. 95% average nucleotide identity clustering) to avoid arbitrary mapping between representatives of highly similar genomes ... The MAGs with the highest relative abundance belonged to the ANME-1 lineage, and together made up >40% of the microbial community at 60 cmbsf (Figure 7 and Supplementary Table 13), with three distinct ANME-1 species (S3_bin4, Co_bin174 and S3_bin12) being particularly abundant (27.47%, 7.33% and 4.69%,

respectively) ... Bacterial liquid alkane and aromatic hydrocarbon degraders were observed in higher relative abundance mainly in sulfate-depleted sediments (1.59-2.69% in 100-250 cmbsf vs 0-0.60% in 0-60 cmbsf sediments; Figure 7 and Supplementary Table 13), consistent with most MAGs lacking genes for respiration (Supplementary Table 5) ... Similarly, *Dehalococcoidia* Co_bin289 and Co_bin57 showed high replication rates (1.3-2.1) in sulfate-depleted sediments, suggesting a syntrophic coupling of long chain alkane degradation by members of Chloroflexi with methanogenesis (Supplementary Table 13).”

End of Discussion L440-464: “Combined metagenomics and metabolomics also suggest that a range of electron acceptors support hydrocarbon degradation in cold seeps. Depth profiles showed that bacteria capable of degrading liquid alkane and aromatic hydrocarbons were mainly detected in sulfate-depleted sediments, suggesting that they mediated hydrocarbon metabolism as part of methanogenic alkane-degrading consortia^{8, 44, 45} ... Thus, despite sulfate availability being thought to limit gaseous alkane oxidation, other electron acceptors may substitute. This advances our understanding of the capacity and mechanisms of anaerobic methane, ethane, propane, and butane oxidation in archaea^{17, 19, 40, 57}.”

The key point of this study – its claim to fame - is to show how different microbial communities or consortia [“the redox-stratified subseafloor microbiome“ of the title] are taking turns in degrading and oxidizing hydrocarbons, depending on sediment depth and electron acceptor availability. Relevant pieces of information are distributed throughout the text or implied in various figures [for example, depth profiles of phylum-level lineages (Fig. 1), key functional genes (Fig. 2a) and key metabolites (Fig. 5) are shown], but there is no summary or synthesis figure to systematically visualize this key message.

It should be possible to look at the data afresh and to extract the information on hydrocarbon-degrading organisms and pathways, and their stratification in geochemical context, to design a strong figure that will summarize and wrap up the manuscript. Instead, the discussion ends with a mention of microbial necromass degradation [line 491]; yes, it exists, but wasn't this manuscript supposed to be about something else? Highlighting and actually showing the central message needs more attention.

Response: This is excellent feedback. We created a summary Figure (revised Figure 7) to summarize key message as suggested. We also removed or consolidated contents unrelated to central message, including most components in Discussion (e.g., necromass, as well as long descriptions of general metabolism) in order to keep the story on point. Additionally, we moved “Metabolomic profiling suggests the community is supported by diverse energy conservation and carbon acquisition strategies” to Supplementary Materials and now only briefly summarized the overall functional capacity of the community in the main text. A summary of the changes made throughout

the manuscript, including its new title, in response to this suggestion from the reviewer, are as follows:

Title: “Depth stratification of the sediment microbiome and thermogenic hydrocarbon degradation at a Scotian Basin seep”

Abstract L39-44: “...Depth distributions of hydrocarbon-oxidizing archaea revealed that they are not necessarily associated with sulfate reduction, which is especially surprising for anaerobic ethane and butane oxidizers. Overall, these findings link subsurface microbiomes to various biochemical mechanisms for the anaerobic degradation of deeply-sourced thermogenic hydrocarbons.”

Revised Figure 7

L197-216: “We further linked the structure of microbial communities to their metabolic capabilities in carbon acquisition and energy conservation strategies ... Metabolomic analysis (Figure 3b) was also performed to identify signature metabolites for anaerobic hydrocarbon biodegradation^{18, 39}.”

Revised Supplementary Note 1

Discussion L405-464: “Biodegradation by sedimentary microbial communities is an important mechanism that controls natural emissions of hydrocarbons from the deep subsurface ... This cultivation-independent genomic investigation of *in situ* seabed biogeochemistry points to a broader role for archaeal multi-carbon alkane oxidation than previously suggested ... Combined metagenomics and metabolomics also suggest that a range of electron acceptors support hydrocarbon degradation in cold seeps ... Thus, despite sulfate availability being thought to limit gaseous alkane oxidation, other electron acceptors may substitute. This advances our understanding of the capacity and mechanisms of anaerobic methane, ethane, propane, and butane oxidation in archaea^{17, 19, 40, 57}.”

Line 387: “both of which are known”

Line 432: “Anaerobic methane oxidizers...”

Response: corrected as suggested.

Changes made:

Line 387 was moved to Supplementary Materials. Line 432 is now as Line 429.

Reviewer #2 (Remarks to the Author):

The conclusion that “upward migrated thermogenic hydrocarbons . . . sustain diverse microbial communities” across a redox spectrum is not really a novel conclusion. Most methane seep sediment studies look at changes in the microbial communities in the sulfate rich and sulfate-depleted depths. Many of these studies have taken place in locations with a significant amount of input from thermogenic hydrocarbon, such as Guaymas Basin and the Gulf of Mexico. I think it is valuable to have another such study, especially in a new location. So, I support the publication of this work. But, it doesn’t strike me as a very novel approach. Perhaps a more compelling rationale would be emphasizing what makes this location different than the others that have been examined.

Response: In response to the reviewer’s important suggestion, we now emphasize our cold seep discovery in a new location (i.e. the NW Atlantic deep sea) and compare this location to the other sites with thermogenic hydrocarbon input that have dominated this research field up until now. We present 3D seismic reflection data to highlight the geology of our study site, and furthermore discuss other differences in temperature and salinity that differentiate our study from previous work.

In this manuscript, which now combines geophysical, geochemical and metabolomic analyses with gene- and genome-centric metagenomics, we successfully link the structure of microbial communities to their metabolic capabilities, discover novel microorganisms related to anaerobic hydrocarbon degradation, and reveal distributions of anaerobic hydrocarbon degraders at different depths. These hydrocarbon oxidizers are not necessarily associated with sulfate reduction, which is especially surprising in the case of anaerobic ethane- and butane-oxidizing archaea.

To further showcase the novelty of this work, we (1) revised the title and Abstract; (2) reanalyzed metabolomic and metagenomic data and revised Results to highlight depth distributions and *in situ* replication rates; (3) revised Discussion to explain the ecological perspective of these findings; (4) added geophysical results describing the new study area.

Changes made:

Title: “Depth stratification of the sediment microbiome and thermogenic hydrocarbon degradation at a Scotian Basin seep”

Abstract L39-44: “...Depth distributions of hydrocarbon-oxidizing archaea revealed that they are not necessarily associated with sulfate reduction, which is especially surprising for anaerobic ethane and butane oxidizers. Overall, these findings link seafloor microbiomes to various biochemical mechanisms for the anaerobic degradation of deeply-sourced thermogenic hydrocarbons.”

Introduction L77-91: “Despite this progress, it remains uncertain whether anaerobic hydrocarbon-degrading isolates or consortia studied in enrichment cultures play these roles *in situ* in deep sea sediments ... However, studies integrating geochemical processes and microbial metabolism in redox-stratified deep sea sediments are lacking.”

Introduction L92-112: “In contrast to hydrothermal sediments, there have been fewer reports on the metabolism of hydrocarbons and other compounds in cold seep sediments, especially in the deep sea ... the microbiome catalysing anaerobic hydrocarbon degradation at different depths is dependent on metabolic adaptations for different redox regimes.”

Results L116-119: “The 3D seismic survey indicated that this site is located above a buried salt diapir (Figure 1b). An overlying seismic amplitude anomaly was interpreted to be a direct hydrocarbon indicator with salt diapir-associated crestal faults suggestive of a potential conduit for fluid migration to the seafloor.”

Revised Figures 1b, 2b, 2c, 3b and 7

New Results sub-section L321-403 - Depth distributions and *in situ* replication rates of hydrocarbon-oxidizing organisms: “To put the metabolic functions of anaerobic hydrocarbon degraders into ecological perspective, relative abundances and cell replication rates were assessed in different sediment depths. The 376 bacterial and archaeal genomes were dereplicated at the species level (i.e. 95% average nucleotide identity clustering) to avoid arbitrary mapping between representatives of highly similar genomes ... The MAGs with the highest relative abundance belonged to the ANME-1 lineage, and together made up >40% of the microbial community at 60 cmbsf (Figure 7 and Supplementary Table 13), with three distinct ANME-1 species (S3_bin4, Co_bin174 and S3_bin12) being particularly abundant (27.47%, 7.33% and 4.69%, respectively) ... Bacterial liquid alkane and aromatic hydrocarbon degraders were observed in higher relative abundance mainly in sulfate-depleted sediments (1.59-2.69% in 100-250 cmbsf vs 0-0.60% in 0-60 cmbsf sediments; Figure 7 and Supplementary Table 13), consistent with most MAGs lacking genes for respiration (Supplementary Table 5) ... Similarly, *Dehalococcoidia* Co_bin289 and Co_bin57 showed high replication rates (1.3-2.1) in sulfate-depleted sediments, suggesting a syntrophic coupling of long chain alkane degradation by members of Chloroflexi with methanogenesis (Supplementary Table 13).”

Discussion L405-464: “Biodegradation by sedimentary microbial communities is an important mechanism that controls natural emissions of hydrocarbons from the deep subsurface ... This cultivation-independent genomic investigation of *in situ* seabed biogeochemistry points to a broader role for archaeal multi-carbon alkane oxidation than previously suggested ... Combined metagenomics and metabolomics also suggest that a range of electron acceptors support hydrocarbon degradation in cold seeps ... Thus, despite sulfate availability being thought to limit gaseous alkane

oxidation, other electron acceptors may substitute. This advances our understanding of the capacity and mechanisms of anaerobic methane, ethane, propane, and butane oxidation in archaea^{17, 19, 40, 57} .”

As far as I can tell, downcore profiles were not made of methane, only of sulfate. The authors should make it clear that they are assuming the presence of a sulfate-methane transition zone, based on the depth where sulfate is depleted. Of course, there is methane in the bottom of the core, so it's possible there was a sulfate-methane transition zone. However, if this sample was in an active methane seep, which wouldn't be apparent with a gravity core, but is a clear possibility, then methane would be present at all layers, and there would be no sulfate-methane transition zone at all. So, the authors cannot say for sure that they have a sulfate-methane transition zone with no downcore methane profile.

Response: Thank you for suggesting this clarification. We removed the classification for biogeochemical zones in the manuscript. We now refer to sediment layers in the study as being sulfate-rich or sulfate-depleted. Additionally, based on sulfate concentrations and sequencing information at 60 cmbsf, we suggest that this represents part of sulfate methane transition zone.

Changes made:

Table 2 and other parts related to this.

L172-175: “ANME-1 (*Methanomicrobia*) comprised 97% of archaea at 60 cmbsf, consistent with the presence of its previously-observed syntrophic partner SEEP-SRB1 bacteria¹², suggesting that this depth was part of sulfate methane transition zone (Figures 2b and 2c).”

L348-352: “In agreement with analysis of 16S rRNA gene fragments in metagenomic libraries (Figure 2c), ANME-1 were also present in deeper sulfate-depleted sediments, with cumulative relative abundances ranging from 0.57% to 8.18% (Figure 7). The cell replication rates calculated using iRep⁵¹ suggested that ANME-1 were still active in these deeper sediments.”

The ANME-1 and GoM Arc 1 dominance seems similar to that of Lloyd et al., 2006 AEM. Perhaps the conclusions of that study should be compared to those of the current study.

Response: Thanks for suggesting this comparison, which has now been included. The main difference between the two studies is the different range of salinity. The 2006 paper described sediments overlying a brine pool methane seep in the Gulf of Mexico whereas in our study the down-core samples show typical marine salinity.

Changes made:

L408-412: “Located on the poorly-explored Scotian Basin in the NW Atlantic, this seep is associated with salt tectonics, similar to other oil- and gas-influenced seabed habitats^{8, 13}. Unlike hypersaline samples in brine seeps such as from the Gulf of Mexico^{13, 53}, this Scotian Basin sediments has a similar salinity to seawater in the cold deep sea.”

L434-439: “Previous observations of these phenomena and enrichment of the corresponding microorganisms have focused mainly on hydrothermally heated sediments^{15, 27, 40} and hypersaline methane seeps^{53, 54}. The results presented here therefore point to a widespread significance for these microbial groups in submarine carbon cycling throughout the global ocean, regardless of salinity and temperature.”

It would be helpful to have a more thorough discussion of whether (and why) the metabolites are considered to be metabolic intermediates, metabolic products, or products of the thermal cracking.

Response: We added an explanation of the metabolomic analysis and revised the caption of metabolite heatmap to highlight other possible sources of these compounds. Alkyl-/arylalkylsuccinates and the newly identified compound ethyl CoM are considered signature metabolites for anaerobic hydrocarbon degradation, whereas other compounds can be derived from multiple biological or abiotic processes.

Changes made:

L214-216: “Metabolomic analysis (Figure 3b) was also performed to identify signature metabolites for anaerobic hydrocarbon biodegradation^{18, 39}.”

Revised Figure 3b and its caption: “Alkyl-/arylalkylsuccinates and ethyl CoM are signature metabolites for anaerobic hydrocarbon degradation, whereas other compounds shown can be derived from multiple biological or abiotic processes.”

Reviewers' Comments:

Reviewer #1:

Remarks to the Author:

My criticisms and comments about the first version of this manuscript have been fully addressed by this revision. By adding depth profiles of physiologically annotated MAGs (Fig. 7), the downcode distribution of different metabolic types [methane oxidation, sulfate reduction, alkane oxidation, methanogenesis) becomes clear.

Details:

Line 180: "...part of the sulfate-methane transition zone..."

Line 330: "... related to the benzoyl-CoA degradation pathway..."

Line 422: " this Scotian Basin sediment..."

The bioRxiv Ref. 55 could be supplemented by a fully published study:

Lloyd, K. G., M. Alperin, and A. Teske. 2011. Environmental evidence for net methane production and oxidation in putative Anaerobic MEthanotrophic (ANME) archaea. *Environmental Microbiology* 13:2548-2564

Figure 2c. Are ANME-3 methane oxidizers reliably differentiated from closely related methanogens? Of all ANME clusters, ANME-3 has the shortest sequence distance to its methanogenic neighbors, and is most susceptible to blurring these phylogenetic boundaries. ANME-1 is easy to identify as a separate family-level lineage, the different ANME-2 lineages are also sufficiently distinct to be reliably identified [approx. genus level}, but ANME-3 is so closely related [2-3 % by 16S rRNA gene sequence] to neighboring members of the Methanosarcinales that hiccups in automated phylogeny annotations are hard to avoid. A control phylogeny may be the best option to check.

Ca. *Syntrophoarchaeum* [red in the color legend] is not really visible in the bar diagram.

Reviewer #2:

Remarks to the Author:

Everything looks great!

Reviewer #1 (Remarks to the Author):

My criticisms and comments about the first version of this manuscript have been fully addressed by this revision. By adding depth profiles of physiologically annotated MAGs (Fig. 7), the downcode distribution of different metabolic types [methane oxidation, sulfate reduction, alkane oxidation, methanogenesis) becomes clear.

Details:

Line 180: "...part of the sulfate-methane transition zone..."

Line 330: "... related to the benzoyl-CoA degradation pathway..."

Line 422: " this Scotian Basin sediment..."

Response: We have corrected them as suggested.

The bioRxiv Ref. 55 could be supplemented by a fully published study:
Lloyd, K. G., M. Alperin, and A. Teske. 2011. Environmental evidence for net methane production and oxidation in putative Anaerobic MEthanotrophic (ANME) archaea. *Environmental Microbiology* 13:2548-2564

Response: We have added the reference as suggested.

Figure 2c. Are ANME-3 methane oxidizers reliably differentiated from closely related methanogens? Of all ANME clusters, ANME-3 has the shortest sequence distance to its methanogenic neighbors, and is most susceptible to blurring these phylogenetic boundaries. ANME-1 is easy to identify as a separate family-level lineage, the different ANME-2 lineages are also sufficiently distinct to be reliably identified [approx. genus level}, but ANME-3 is so closely related [2-3 % by 16S rRNA gene sequence] to neighboring members of the Methanosarcinales that hiccups in automated phylogeny annotations are hard to avoid. A control phylogeny may be the best option to check.

Response: Thanks for this information. Figure 2c is not based on phylogeny annotations. Their relative abundances were calculated using phyloFlash, which summarized taxonomic diversity of a metagenome library from SSU rRNA read affiliations. This process is based on direct read mapping to the SILVA database (release 132). To make it clear, we added an explanation in the legend: "...Top panels: 16S rRNA gene fragments derived from metagenomic libraries using the phyloFlash pipeline..."

Ca. Syntrophoarchaeum [red in the color legend] is not really visible in the bar diagram.

Response: As *Ca. Syntrophoarchaeum* is much lower compared to others, it is not visible in the bar diagram. We removed this color key in order not to confuse readers.

Reviewer #2 (Remarks to the Author):

Everything looks great!

Response: Thank you very much for your feedback.